# KALM: Knowledgeable Agent by Offline Reinforcement Learning from Large Language Model Rollouts

**Jing-Cheng Pang, Si-Hang Yang, Kaiyuan Li, Xiong-Hui Chen, Nan Tang, Yang Yu**[*]
National Key Laboratory for Novel Software Technology, Nanjing University, China
& School of Artificial Intelligence, Nanjing University, China
Polixir Technology
{pangjc,yangsh,liky,chenxh,tangn}@lamda.nju.edu.cn, yuy@nju.edu.cn

## Abstract

Reinforcement learning (RL) traditionally trains agents using interaction data, which limits their capabilities to the scope of the training data. To create more knowledgeable agents, leveraging knowledge from large language models (LLMs) has shown a promising way. Despite various attempts to combine LLMs with RL, there is commonly a semantic gap between action signals and LLM tokens, which hinders their integration. This paper introduces a novel approach, KALM (Knowledgeable Agents from Language Model Rollouts), to learn knowledgeable agents by bridging this gap. KALM extracts knowledge from LLMs in the form of imaginary rollouts, which agents can learn through offline RL. To overcome the limitation that LLMs are inherently text-based and may be incompatible with numerical environmental data, KALM fine-tunes the LLM to perform bidirectional translation between textual goals and rollouts. This process enables the LLM to understand the environment better, facilitating the generation of meaningful rollouts. Experiments on robotic manipulation tasks demonstrate that KALM allows agents to rephrase complex goals and tackle novel tasks requiring new optimal behaviors. KALM achieves a 46% success rate in completing 1400 various novel goals, significantly outperforming the 26% success rate of baseline methods. Project homepage: https://kalmneurips2024.github.io.

## 1 Introduction

Developing knowledgeable agents that complete diverse manipulation tasks is a hallmark of machine intelligence. Reinforcement learning (RL) has emerged as a powerful mechanism for training intelligent agents to acquire such abilities in interactive environments [1, 2, 3]. In this approach, agents can learn from the environmental interaction data. Although RL has shown promise in many challenging tasks, it is limited by the scope of the available interaction data: agents often struggle to accomplish tasks not covered by the interaction data and lack the ability to generalize to new or slightly altered tasks. For instance, a policy trained to *move the block to the left* often fails to complete the instruction *move the block to the right*, despite them being highly similar tasks.

On the other hand, advancements in large language models (LLMs) have opened up new opportunities for developing intelligent agents that solve general tasks in text domain [4, 5, 6, 7, 8, 9]. Trained on extensive text corpora, LLMs embed a wide array of general world knowledge. Developing approaches effectively leveraging such knowledge to build knowledgeable agents for interactive and physical tasks recently presents a promising research frontier [10]. Fortunately, existing research evidence has indicated LLMs' utility beyond textual domains. They successfully utilize LLMs

---

[*]Corresponding: yuy@nju.edu.cn

38th Conference on Neural Information Processing Systems (NeurIPS 2024).

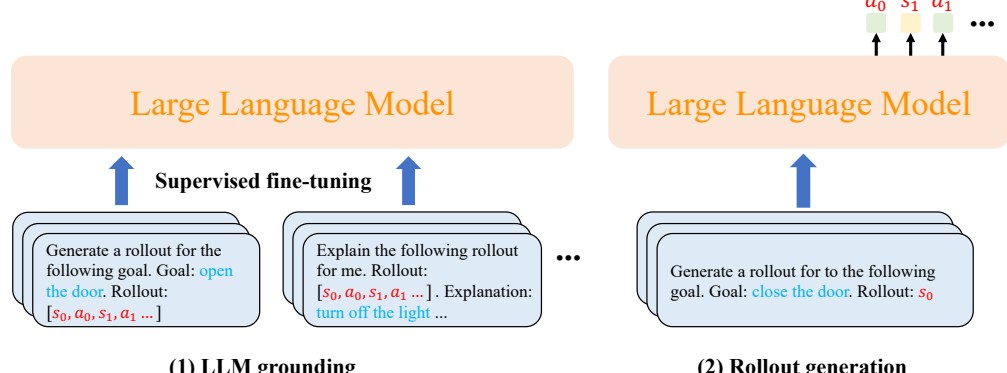

Figure 1: Illustration of KALM utilizing LLM to generate environmental rollouts at the numerical vector level. **(1)** Grounding phase that fine-tunes LLM with supervised fine-tuning on the environmental data. **(2)** Generation phase that prompts LLM to generate data for novel skills. KALM modifies the input/output layer of LLM, enabling it to process and interpret non-textual data.

to decompose complex tasks and make high-level decisions in interactive, physical environments [11, 12, 13]. However, they are constrained by a set of available skills and gap still exists between LLMs and the environments: the LLMs are inherently text-based, while the physical environments often operate with numerical data. As a result, previous methods with LLMs tend to focus on high-level, text-based decision-making and do not adequately address low-level control, especially when handling numerical vector inputs and outputs.

This study investigates developing knowledgeable agents with RL and LLMs, which achieve low-level control and adapt to novel situations. We introduce KALM (Knowledgeable Agent from Language Model rollouts) method, which employs a pre-trained LLM to create imaginary rollouts for novel skills, which agent can easily learn through offline RL techniques. The motivation behind KALM is that LLMs, with their extensive knowledge repository, are ideally suited for generating sequences that simulate the completion of novel goals. However, a key challenge is that LLMs are inherently designed to process textual data, while the environmental data is often in numerical vectors. To address this, KALM applies two techniques to ground LLM and process the environmental data: (1) adapting the LLM's architecture to handle environmental states and actions, and (2) fine-tuning the LLM in a supervised manner, e.g., predict execution rollouts and translate between natural language goals and their corresponding rollouts. Subsequently, KALM uses the LLM to generate imaginary rollouts for various goals, including rephrasings of existing goals and completely novel tasks not present in the interaction data. Fig. 1 depicts the process by which KALM utilizes an LLM to generate imaginary rollouts. Finally, KALM integrates offline RL techniques to acquire novel skills. This novel combination of LLM-generated imaginary rollouts with offline RL potentially yields more versatile and knowledgeable agents.

Our contributions are as follows: We introduce an effective way to integrate RL and LLMs for low-level control, enhancing the RL training with LLM knowledge through imaginary rollouts. Unlike previous works that primarily leverage LLM for text-based high-level control, we demonstrate that LLM knowledge can be utilized in interactive environments that operate on numerical vectors. Additionally, we present a technique for effectively aligning LLMs with environments, enabling LLMs to comprehend environmental data from different modalities. Lastly, we verify the efficacy of KALM on two robotics manipulation environments: CLEVR-Robot [14] and Meta-world [15]. Experiment results show that KALM successfully fine-tunes the LLM to generate meaningful rollouts. In the CLEVR-Robot simulation environment, it achieves a 46% success rate on tasks with 1400 various novel goals, significantly outperforming the baseline offline RL method's 26% success rate.

## 2 Background and Related Work

### 2.1 Background

**Reinforcement learning.** We consider an RL problem where the agent completes goals assigned by natural language. We model the environment as a goal-augmented Markov Decision Process

(MDP) [16, 17, 18], represented by the tuple $\mathcal{M} = (\mathcal{S}, \mathcal{A}, \mathcal{P}, \mathcal{R}, \gamma, \mathcal{G})$. In this tuple, $\mathcal{S}$ denotes the state space, $\mathcal{A}$ the action space, $\mathcal{P}$ the transition function of the environment, $\mathcal{R}$ the reward function that evaluates the quality of the agent's action, $\gamma$ the discount factor which balances the immediate and future rewards, and $\mathcal{G}$ the set of natural language goals. A policy $\pi : \mathcal{S} \times \mathcal{G} \rightarrow \Delta(\mathcal{A})$ defines the agent's strategy, mapping states and goals to a distribution over possible actions. The interaction between the RL agent and the environment proceeds as follows: at each timestep $t$, the agent observes a state $s_t$ and a goal $G$ from the environment. It then selects an action $a_t$ based on the policy $\pi(\cdot|s_t, G)$. Upon executing this action, the agent receives a reward $r(s_t, a_t, G)$ and the environment transitions to a new state $s_{t+1}$ according to the transition function $\mathcal{P}(\cdot|s_t, a_t)$. The objective of RL is to find a policy that maximizes the expected sum of rewards over time. In this study, we call the state and action data of the environment the *environmental data*.

**Large language model.** LLM refers to an autoregressive text generation model that predicts future tokens in a sequence, where each token is predicted as $l_{t+1} = \mathcal{M}(E_T(l_0), \cdots, E_T(l_t))$, conditioned on all prior sequence of tokens. Here, $E_T$ is the token embedding layer that converts the token into a D-dimensional embedding, $l_t \in \Sigma$, and $\Sigma$ denotes the vocabulary of the LLM. Specifically, the token embedding layer converts tokens into embeddings $e_k = E_T(l_k) \in \mathbb{R}^D$, and the output layer of a LLM classifies tokens. In this study, we consider LLM, which operates on textual data, environmental state, and action data. To process state and action data, we modify the LLM's architecture by replacing its original input and output layers with additional multi-layer perceptrons (MLPs), thereby enabling the integration of non-textual environmental data.

## 2.2 Related Work

**Offline Reinforcement Learning.** Offline RL [19, 20, 21] enables agents to learn from a static dataset of pre-collected experiences without real-time environment interaction. The core challenge in offline RL is to derive effective policies from a dataset that may be biased or has limited data. Behavior cloning [22] is a straightforward solution by mimicking the behavior present in the dataset. Besides, prior studies have introduced novel techniques such as importance sampling, conservative policy evaluation, and representation learning to address these challenges [23, 24, 25, 26]. Trajectory transformer (TT) [27] treats offline RL as a sequence modelling problem, aiming to produce a sequence of actions that leads to high rewards. Unlike TT, KALM utilizes a pre-trained LLM to generate rollouts instead of as a policy. Despite offline RL's successes, it is limited by the diversity of the dataset: if the dataset lacks specific experiences, the agent may fail to perform adequately in those scenarios. Model-based RL (MBRL) methods [28, 29] learn a dynamic model from offline data, which can then be used by any policy learning algorithm to recover the policy. MOPO [30] and MOReL [31] use uncertainty quantification to construct a lower bound, aiming to avoid issues like model bias and distribution shift. COMBO [32] employs both the offline dataset and model-generated rollouts to train a value function and regularize it on out-of-support state-action tuples. Despite both MBRL and KALM methods utilized generated rollouts for policy training, they are different from motivation. MAPLE [33] and ReDM [34] attempt to overcome this by training multiple diverse environment models to simulate wide range of environmental rollouts, thereby enhancing policy robustness in unfamiliar scenarios. These environment models, however, are typically learned from scratch and prioritize extensive data coverage, which may not align with real-world data distributions. In contrast, KALM leverages a pre-trained LLM to facilitate the generation of imaginary rollouts, taking advantage of the general knowledge from the pre-trained LLMs.

**Large Language Models.** Large language models (LLMs) exhibit remarkable proficiency in processing and comprehending natural language [35, 36, 37, 38]. More exciting, their capabilities extend to tasks beyond basic language understanding problems, including dialogue [4, 39, 5], multimodal vision-language tasks [6], logical reasoning [7, 8], and mathematical problem-solving [40]. Recent advancements, such as GPT-4 [35], have pushed the boundaries further, not only in terms of text processing but also in exploring the capabilities of LLMs in interactive environments [41, 42, 43], capitalizing on their embedded knowledge of the world. How to effectively utilize this knowledge for decision-making has emerged as a promising research area [44, 42]. In this work, we propose a novel idea of leveraging the knowledge in LLMs as imaginary rollouts to develop knowledgeable agents.

**LLMs for RL.** A promising area of study is how to effectively leverage LLMs to enhance RL in interactive tasks. Research in this domain has taken several approaches to leverage LLMs. One approach is built on a hierarchical framework, utilizing LLMs to decompose complex tasks and

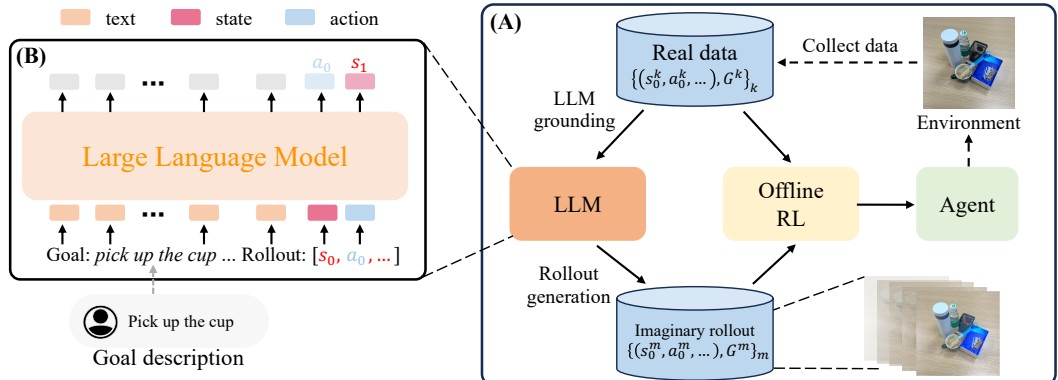

Figure 2: Overall framework of KALM method. **(A)** Overall training procedure, consisting of three key steps: LLM grounding that fine-tunes LLM with environmental data, rollout generation that prompts the LLM to generate imaginary rollouts for novel tasks, and offline RL training that facilitates skill acquisition. The dashed line (- - ->) represents an optional online process, allowing for continuous improvement through iterative data collection and training. **(B)** The adapted network structure of the LLM. KALM adapts the architecture of LLM to process both text and environmental data.

generate high-level plans, which are then executed by a low-level controller [12, 45, 43]. Another approach uses LLMs to design the reward function [46, 47, 48], which streamlines the otherwise laborious process of reward function formulation. Recently, some works have investigated directly employing LLMs as the behaviour policies, training them with RL to facilitate direct interaction with the environments. For example, GALM [41] and TWOSOME [49] ground LLMs in text-based games, and LLaRP [42] trains a LLM to output actions for embodied tasks with RL. Different from these studies utilizing LLMs at the text level or as policies, in this work, we propose a novel approach to utilizing LLM, which extracts knowledge in the form of imaginary rollouts.

## 3 Method

This section presents the proposed Knowledgeable Agents from Language Model Rollouts (KALM) method. Fig. 2 shows the framework of KALM. We first give a formal definition of the problem and then elaborate on the three critical steps of KALM: (1) LLM grounding that enables LLM to understand the elements of the environment, (2) rollout generation that generates imaginary rollouts for novel skills, and (3) Skill acquisition that trains the policy with offline RL.

### 3.1 Problem Formulation

Consider that we have an offline dataset $\mathcal{D}$, which is collected from the environment. This dataset consists of pairs of goals and corresponding interaction rollouts: $\{G^k, (s_0^k, a_0^k, s_1^k, a_1^k, \cdots)\}_{k=1}^K$. In this context, the sequence $(s_0^k, a_0^k, s_1^k, a_1^k, \cdots)$ represents a rollout detailing the sequence of states and actions $(s_i^k, a_i^k)$ required to complete the goal $G^k$. The primary objective here is to obtain a policy that achieves high rewards on unseen goal distributions, represented as $\mathcal{G}'$, thus ensuring its decision-making ability beyond the constraints of the available interaction data. We will define the 'unseen goal distributions' in Sec. 3.3.

### 3.2 LLM Grounding by Supervised Fine-tuning

The first step of KALM is to ground LLM in the environment. The purpose of this module is to enable LLM to interpret the meaning of states, actions, dynamics and rollouts of the given environment. To achieve this, we train the LLM using the dataset $\mathcal{D}$ to perform three different tasks via supervised fine-tuning (SFT):

- Dynamics prediction: The LLM predicts environmental dynamics changes. Given the current state $s_t$ and action $a_t$, the LLM predicts the next state $s_{t+1}$.

- Rollout explanation: The LLM is presented with a rollout sequence $s_0, a_0, s_1, \cdots$, and it is required to describe the rollout with natural language.
- Rollout generation: The LLM generates a rollout that aligns with a specified goal $G$.

We can construct the supervised training data based on the data from $\mathcal{D}$. Here, KALM regards the LLM grounding problem as an instruction-following problem: LLM demonstrates excellent performance following given natural language instructions to generate the desired answer. This way, we can adjust the instruction prompt input to the LLM to better utilize it and specify its generation objective. The instruction prompts we use for SFT are presented in Appendix A.4.

Given that LLMs are initially trained on textual data to process and predict sequences of text tokens, they can not be directly utilized to handle numerical vectors. To overcome this, KALM introduces modifications to the LLM's network architecture. As shown in Fig. 2(B), we use a pre-trained LLM as the backbone model and modify it with additional layers to handle environmental data. For inputs such as states and actions, we incorporate learnable embedding modules, $E_S : \mathcal{S} \to \mathbb{R}^D$ and $E_A : \mathcal{A} \to \mathbb{R}^D$, which transform these inputs into embeddings of the same dimensionality as the token embeddings. For outputs, we employ learnable modules, $O_S : \mathbb{R}^D \to \mathcal{S}$ and $O_A : \mathbb{R}^D \to \mathcal{A}$, which map the LLM's output into state space $\mathcal{S}$ or action space $\mathcal{A}$. This framework can be easily extended to visual observation tasks by integrating appropriate neural network architectures.

### 3.3 Rollout Generation with Goal-oriented Prompt

After fine-tuning with environmental data, the LLM acquires the capability to interpret the states, actions, and dynamics within the environment. Given that LLMs possess a broad spectrum of world knowledge, they have the potential to generate imaginary rollouts for a diverse range of novel skills. To this end, we employ the fine-tuned LLM to generate imaginary rollouts given the initial state $s_0$ and the goal: $\{a_0, s_1, a_1, \cdots\} \leftarrow \mathcal{M}(GOP, s_0)$. Here, $GOP$ stands for *goal-oriented prompt*: "Generate a rollout for the following goal: [GOAL]. Rollout: ", where "[GOAL]" is a placeholder for various goals that reflect different skills. Following prior research that studies policy generalization under natural language goals [42], we measure the novelty of the goal along two dimensions:

- Paraphrastic Robustness: This dimension assesses the agent's consistency in optimal behavior when faced with linguistically diverse goals that share the same underlying intent as previously seen goals. It includes alternative phrasings for identical actions or re-expressing the name of the objects or entities.
- Novel Task Generalization: Here, we investigate the agent's proficiency in performing tasks that require the formulation of new optimal behaviors. Such tasks do not exist in the dataset $\mathcal{D}$. For instance, the dataset includes tasks related to making a robot walk, while a novel task enables the robot to run. To generate effective rollouts for these tasks, the LLM is required to understand the entity's meaning and the environmental dynamics correctly.

We will elaborate on how we construct novel tasks that align with these dimensions in Sec. 4.1.

### 3.4 Offline Reinforcement Learning for Skill Acquisition

KALM employs offline RL approach to train a policy $\pi(\cdot|s, G)$, utilizing both the real and imaginary rollouts generated from LLM, with the same proportion of two sources of rollouts. To build a policy network, BERT [50] serves as the encoder for processing natural language goals due to its proficiency in text encoding. The goal encoding is integrated with the state representations to form the input for the policy network. For policy optimization, KALM is compatible with any offline RL algorithm, such as TD3+BC [51] and CQL [23], leveraging the combined data from the offline dataset and imaginary rollouts. Furthermore, we comprehensively analyse KALM's performance when integrated with different offline RL algorithms in Sec. 4.3.

## 4 Experiment

In this section, we conduct experiments to evaluate the efficacy of the KALM method. We aim to address the following important questions: (1) How does KALM perform on novel goals compared to

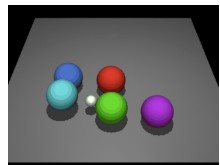 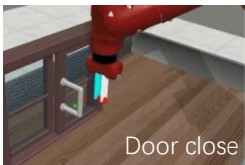 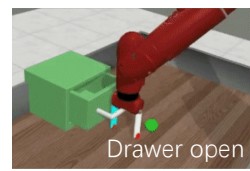 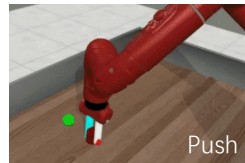

(A) CLEVR-Robot               (B) Meta-world

Figure 3: A visualization of the environments in our experiments. **(A)** In the CLEVR-Robot environment, the agent (silverpoint) manipulates five movable balls to reach a goal configuration. **(B)** In Meta-world, the agent controls a Sawyer robot to manipulate various objects.

existing baseline methods (Sec. 4.2)? (2) How are the rollouts generated by the fine-tuned LLM (Sec. 4.3)? (3) How does KALM ground LLM in the environment (Sec. 4.3)? (4) What is the impact of each component in KALM on the overall performance of the algorithm (Sec. 4.4)? We first introduce the environment used for experiments and the specific settings employed in the evaluation.

## 4.1 Experimental Setting

**Evaluation environments.** We conduct experiments on two benchmark robotics manipulation environments built based on MuJoCo physics engine [52]: Meta-world [15] and CLEVR-Robot [53], as depicted in Fig. 3. In the Meta-world, the agent controls a Sawyer robot to manipulate various objects, e.g., doors, drawers and windows. The target configurations in the offline dataset involve: reach, push, pick-place, button-press, door-unlock, door-open, window-open, faucet-open, coffee-push and coffee-button-press. The state space is defined in $\mathbb{R}^{91}$, representing the robotics arm's state and the different objects' location & orientation. The action space is $\mathbb{R}^4$, denoting the gripper movement and open/close.

In CLEVR-Robot, the agent (silverpoint) manipulates five movable balls. The target configuration in the offline dataset involves *moving a specific ball in a target direction (front, behind, left, or right) relative to a target ball*. The state space is defined in $\mathbb{R}^{10}$, representing the positions of the balls, while the action space is $\mathbb{R}^{40}$, where the action is a one-hot vector representing the movement of a specific ball to a direction.

**Novel task for evaluation.** To evaluate the efficacy of KALM and the learned policy's generalization to novel tasks, we define novel tasks with three levels of complexity as follows:

- Rephrasing goal: The agent performs the same manipulation tasks as offline data but receives paraphrased goals which are not present in the data. For example, the goal in offline data is *move the blue ball to the front of the red ball*, while the paraphrased goal could be *I really dislike how the red ball is positioned in front of the blue ball. Could you exchange their places?*

- Unseen (easy): The agent is tasked with different manipulation tasks that do not exist in the dataset, requiring the LLM to understand the environmental data well. To correctly generate the imaginary rollouts, the LLM must understand the meaning of state, action, dynamics and their relation with the goals.

- Unseen (hard): The agent faces tasks substantially different from those in the offline dataset, which require a complex composition of behaviors, such as "Gather all balls together", and "Move five balls to a straight line" in the CLEVR-Robot environment. For these tasks, LLM needs to understand the meaning of environmental data and create a novel combination of state and action to generate meaningful rollouts.

Due to the space limitation, we present detailed descriptions and examples of novel goals in Tab. 2. Before utilizing LLM to generate rollouts, we prompt ChatGPT [9] to output the natural language goals that describe different novel target configurations (which are not present in the offline dataset). Then, we prompt the fine-tuned LLM to generate imaginary rollouts for various novel goals.

**Dataset collection and rollout generation.** For Meta-world and CLRVR-Robot, the pre-collected offline dataset consists of 100,000 rollout-goal pairs, each corresponding to the state, action, and reward sequences for completing the natural language goal. Here, we employ the environment-built-in reward function to obtain and incorporate rewards for offline datasets and imaginary rollouts. The dataset encompasses 80 unique target configurations. Following prior research [13], we let ChatGPT

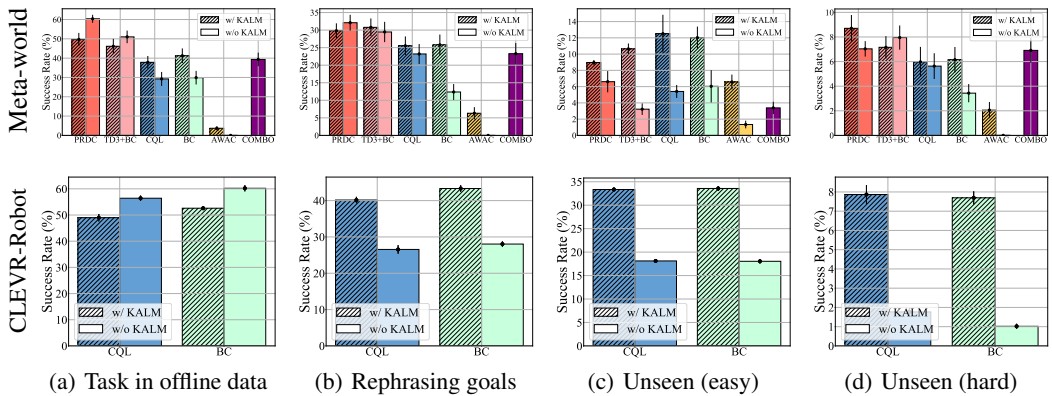

Figure 4: Success rate bars of different methods on various levels of goals. The x-axis denotes the offline RL algorithm, and the y-axis denotes the success rate for completing various natural language goals. The success rate is calculated based on the average of the last five checkpoints, and the error bars stand for the half standard deviation over three random seeds. We present the training curves in Fig. 8.

[9] generate 18 different natural language sentence patterns to describe each target configuration, resulting in 1440 different natural language goals. For the Meta-world, the dataset involves 10 different target configurations, and we employ ChatGPT to generate 20 different natural language goals to describe each target configuration, resulting in 200 different goals in total.

When KALM fine-tunes LLM, we construct a training set comprising 300,000 trajectories, each rollout-goal pair in the offline dataset extending three trajectories: dynamics prediction, rollout explanation, and rollout generation (as detailed in Sec. 3). When training policies with offline RL, baseline methods use the offline data (6400 rollout-goal pairs), while KALM generates additional 5600, 72400 and 1680 imaginary rollouts for rephrasing goal, unseen (easy) and unseen (hard) tasks, respectively. For each level of novel goals, KALM trains the policy on the offline dataset and the generated rollouts at the corresponding level, with the same proportion of offline data and imaginary rollout in each training batch. There are 1400 and 240 novel goals for CLEVR-Robot and Meta-world, respectively.

**Implementation details.** We utilize the Llama-2-7b-chat-hf model [36] as the backbone LLM across the experiments. The LLM undergoes training for 5 epochs with batch size 32 on Meta-world and 10 with batch size 24 on CLEVR-Robot. We implement baseline methods by utilizing d3rlpy [54], a well-established code base. We utilize Adam optimizer [55] to optimize policy. All methods train policy for 500,000 gradient steps. The offline RL training is replicated with three different random seeds to ensure the robustness of the results. Details about hyper-parameters are provided in Appendix A.3. We implement all the above modules using 64 AMD EPYC 9374F 32-Core Processor, 8 NVIDIA GeForce RTX 4090 cards and 1TB RAM.

## 4.2 Main Results

**Baselines for comparison.** We consider representative offline RL methods that train policies on offline data for comparison. We briefly introduce them as follows: (1) PRDC ([56]) is an offline RL method regularizing the policy towards the nearest state-action pair in the offline data based on tree-search method. (2) Behavior Cloning (**BC**) adopts a supervised learning approach to mimic the actions within the offline dataset. (3) Conservative Q-Learning (**CQL**) [23], a prominent offline RL algorithm, constructs a conservative Q-function that ensures the policy's expected value does not exceed its true value. (4) **TD3+BC** [51] guides the agent to stay closer to the demonstrated behavior while benefiting from TD3 [57] algorithm's stability and efficiency. (5) **AWAC** [58] weights the actions according to their estimated advantages to improve policy learning. We use the suffix '+KALM' to denote the method that trains policy on offline datasets and the imaginary rollouts. To ensure a comprehensive evaluation, we also consider a baseline, LLM as a policy that takes advantages of both offline data and LLM. (6) **COMBO** [32] utilizes ensemble environment models to achieve conservative policy learning.

**Comparison with offline RL methods.** Fig. 4 presents the comparative performance on two CLEVR-Robot and Meta-world. In Fig. 4(a), these methods with the 'KALM' suffix denote that the imaginary

rollouts are generated for rephrasing goals task. Overall, these methods equipped with KALM gain a clear improvement over the baseline methods, especially on these tasks with novel goals (see last three columns in the figure). For example, CQL+KALM achieves averaged a success rate of 27.1% on novel tasks in CLEVR-Robot, while the score of CQL is 15.5%. Besides, on task in offline data, KALM outperforms or is comparable to baselines, indicating that the inclusion of generated rollouts can not only preserve the performance but also potentially enhance the performance on these tasks within the real data. This could be attributed to the fact the imaginary rollouts improve the diversity of the dataset. It is worth noting that baselines show an clear decrease in performance on the rephrasing goal task compared to the performance on the task in offline data. This indicates that policies trained exclusively on offline data exhibit limited generalizability to rephrasing the goals they have encountered. In contrast, policies incorporating imaginary rollouts exhibit remarkable performance enhancements over baselines (see results on rephrasing goals and unseen (easy) tasks). When the task becomes more novel and complex, the performance improvement brought by KALM gets more significant in improving proportion. In comparison, baseline methods hardly progress on unseen (hard) tasks, reinforcing the utility of generated rollouts in acquiring complex novel skills. While COMBO gets considerable score on seen tasks and rephrasing goals, it performs poorly on unseen (easy) and unseen (hard). This can be attributed to that their environment models are learned from scratch, lacking the ability to generalize novel goals and leading to the low scores on unseen tasks.

**LLM as a direct policy.** We also consider a baseline that utilizes offline data and LLM, which we call 'LLM as policy'. In this experiment, the fine-tuned LLM directly outputs the action given the goal-oriented prompt GOP (refer to Sec. 3.3) and previous states and actions: $a_t = \mathcal{M}(\text{GOP}, s_0, a_0, \cdots, s_t)$. We also consider DT [59] that utilizes both LLM and offline data, with Llama-2-7b-chat-hf as the backbone policy model. DT treats decision-making as a sequence modeling problem, using a transformer architecture to predict actions based on the desired future return. We evaluate the baseline on CLEVR-Robot, with the results presented in Fig. 5. The report KALM results are success rate evaluated with CQL+KALM policy. We observe that KALM outperforms both LLM as policy and DT.

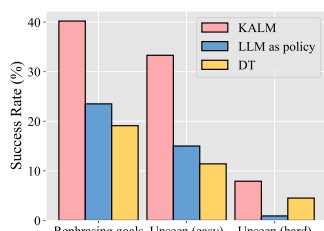

Figure 5: Comparison with baseline method that directly utilizes LLM as policy.

LLM as policy yields a relatively low success rate. This result may be due to the primary focus of the LLM's fine-tuning process, which is to capture sequence information related to completing the given goal rather than to precisely predict actions, as trajectory transformers do [27]. While the action might be sub-optimal, the generated rollouts can still contribute positively to skill acquisition in offline RL. The comparison results with LLM as policy method suggest that the current approach to simultaneously modelling state and action sequences may lead to inaccurate behaviours. Future work should explore the possibility of separating the modelling of states and actions by employing two distinct LLMs, thereby enhancing the accuracy of behaviour prediction in complex environments.

## 4.3 Performance of LLM Grounding

The previous section demonstrates that KALM improves policy's performance on unseen novel goals. In this section, we dive into the details of the KALM running process and examine why KALM can improve the policy performance. Specifically, we demonstrate the examples of the imaginary rollouts and the accuracy of the rollout explanation, which reflects the performance of LLM grounding.

**Analysis on the LLM rollouts**. To investigate the quality of the generated rollouts, we showcase illustrative examples of the imaginary rollouts in Fig. 6. We reset the environment to the generated state to obtain the visualization image. In the meta-world environment, the LLM generates a rollout to reach the target point behind a wall. Although the object 'wall' never occurs in the offline data, the LLM can adjust the robotics arm's trajectory to avoid collision. This adaptation underscores the LLM's comprehension of the environment and ability to leverage prior knowledge. For CLEVR-Robot, the goal 'gather all balls close to green' deviates significantly from the goals in the offline dataset. Notably, the LLM correctly identifies the green ball and orchestrates the movement of the remaining balls towards this target. These results support using pre-trained LLM to generate imaginary rollouts for novel goals. We present more examples of the generated rollouts and corresponding analysis in Appendix C.2.

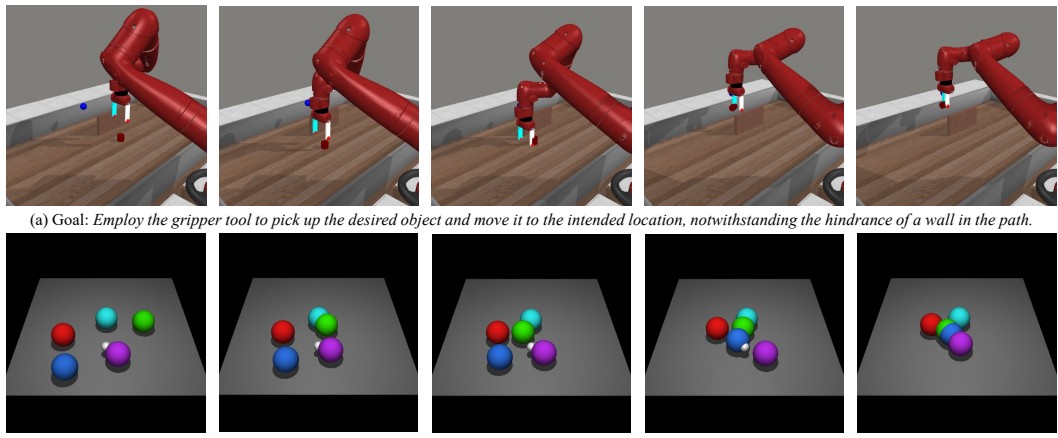

(a) Goal: *Employ the gripper tool to pick up the desired object and move it to the intended location, notwithstanding the hindrance of a wall in the path.*

(b) Goal: *Use the green ball as the nucleus of the circle, arranging the rest around it.*

Figure 6: Examples of the imaginary rollouts generated by the fine-tuned LLM.

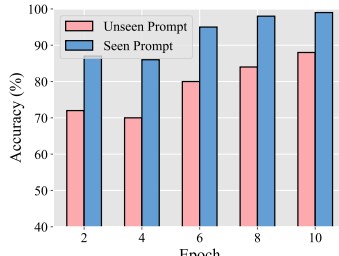

Figure 7: Rollout explanation accuracy.

| Task
Method | Rephrasing goals | Unseen (easy) | Unseen (hard) |
|---|---|---|---|
| w/o rollout explanation | 83% | 39% | 8% |
| w/o dynamics prediction | 83% | **42%** | 15% |
| KALM | **85%** | 36% | **28%** |

Table 1: Ablation study on the training objective of the LLM fine-tuning. The values represent the match rate of the generated rollouts with the given goals. Results indicate that the rollout explanation term has a deeper impact on the generation.

**Accuracy of the rollout explanation**. We measure the LLM ground in terms of LLM's ability to explain environmental rollouts, where the LLM is asked to explain the rollouts with both seen prompt (Appendix A.4) and unseen prompt (*Suppose you are playing gaming of five balls with different colours. Please explain the following rollout briefly.\n rollout: [ROLLOUT]\n Answer:[ANSWER]*). The accuracy of the explanation is calculated by the keyword match between the LLM output and the ground-truth label. As depicted in Fig. 7, the LLM demonstrates a high level of explanatory accuracy, even after only two training epochs. These results suggest that the LLM correctly captures the meaning of rollouts and retains natural language expression capability after fine-tuning. Besides, after supervised fine-tuning, the rollout explanation accuracy exceeds 85% on unseen prompts, demonstrating that the LLaMA2 model retains its language understanding capability.

## 4.4 Ablation Study

We conduct ablation evaluation on KALM with different SFT objectives, i.e., dynamics prediction and rollout explanation (refer to Sec. 3.2). We maintain consistent hyper-parameters across experiments to isolate the effects of these training objectives. The match rate between the generated rollouts and the goals measures the effectiveness of the training. Tab. 1 illustrates that the rollout explanation objective is more critical than dynamics prediction. This is attributed to the fact that the rollout explanation assists the LLAMA2 in capturing the temporal consistency between the rollout sequences and corresponding goal descriptions. Besides, integrating all three training objectives has yet to improve on unseen (easy) tasks. The performance degradation is attributed to the specific nature of the unseen (easy) task, whose objective is to predict one-step transitions given unseen language goals. To be more specifically, we would discuss the two components of LLM fine-tuning in KALM (i.e., dynamics prediction and rollout explanation) respectively. For dynamics prediction objective, unseen (easy) task objective (predicting and given and ) shares similarity, yet diverges from the dynamics prediction objective (predict given and ). This difference introduces a potential conflict in the LLM SFT. This is evidenced by the empirical results that KALM w/o dynamics prediction achieves highest rollout accuracy for unseen (easy) task. For rollout explanation objective, it focuses on giving explanation on a long rollout sequences. While this objective enriches the model's capability to

provide coherence over temporal sequence, it may inadvertently detract from the model's ability to capture the immediate logic of transitions between adjacent two steps.

## 5    Conclusion and Limitation

This study investigates the integration of RL and LLMs for low-level control in interactive, physical environments. We introduce a novel method, KALM, which bridges the gap between LLMs and environments, and extracts the knowledge from LLM in the form of imaginary rollouts. Offline RL is then applied to facilitate skill acquisition from the imaginary rollouts. The experiment results on two robotics manipulation tasks validate the effectiveness of KALM and demonstrate the feasibility of acquiring knowledge from language model rollouts. However, there are still some limitations. First, the LLM learns to generate both state and action, while the action is correlated with the behavior policy. This dual responsibility increases the burden on the LLM, requiring it to learn the environment and imitate the behavior policy simultaneously. A solution could be employing two LLMs to learn state and action modelling separately. Second, current experiments only consider the state in the form of vector. Future work can evaluate KALM on tasks with state in other modalities, e.g., image, incorporating a vision encoder to handle visual data. Last, all parameters of LLM trained during the grounding phase, potentially influencing the embedded knowledge and hindering ability of the LLM to generalize novel goals. Incorporating a general text dataset during fine-tuning could mitigate this issue. We believe these interesting directions are worth further exploration for developing smarter and more robust agents with the support of large pre-trained models.

## Acknowledgment

This work is supported by Jiangsu Science Foundation (BK20243039). The authors extend their appreciation to Chengxing Jia and Yidi Wang for their detailed discussions on the implementation details, and anonymous reviewers for providing valuable comments during the reviewing process.

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

# Appendix

## Table of Contents

# A Experiment Details

## A.1 Detailed Descriptions of Natural Language Goals for Two Environments

| Level of novelty | CLEVR-Robot | Meta-world |
|---|---|---|
| Tasks in dataset | Move a specific goal to one direction of the target ball. Examples: (1) *Can you push the red ball to the left of the blue ball? (2) Help me keep the green ball behind the purple ball. (3) Move the cyan ball in front of the red ball. (4) Can you help me move the blue ball to the right of the red ball?* | Agent completes ten different tasks (provided by Meta-world benchmark), including Reach, Push, Pick-Place, Button-Press, Door-Unlock, Door-Open, Window-Open, Faucet-Open, Coffee-Push, Coffee-Button-Press. Examples: (1) Reach: *Adjust the gripper to the desired location.* (2) Coffee-Button-Press: *Utilize the gripper to depress the button on the coffee machine.* |
| Rephrasing Goal | The task goals involve moving a specific goal to one direction of the target ball, but in different expressions as in the offline dataset. Refer to A.2.1 for rephrasing examples of natural language goals. | Agent completes ten tasks in the dataset with the natural language expressions rephrasing from the tasks in the dataset. Example: (1) Push: *The target object's present placement does not meet my preferences; kindly use the gripper to push it to the desired position.* (2) Window-Open: *The closed window bothers me; could you please operate the gripper to open it?* |
| Unseen (Easy) | Agent moves a specific ball in a given direction in one step. This task does not exist in the offline dataset but requires the agent's deep understanding of the natural language goal. Refer to A.2.2 for unseen-easy examples of natural language goals. | Agent completes eight novel tasks, including Reach-Wall, Push-Wall, Pick-Place-Wall, Button-Press-Wall, Door-Lock, Door-Close, Window-Close, Faucet-Close. Example: (1) Button-Press-Wall: *Apply the gripper to activate the button, yet find its path obstructed by an unexpected wall.* (2) Door-Close: *Employ the gripper to shut the door.* |
| Unseen (Hard) | Agent completes tasks substantially different from original tasks, such as "gather all the balls together", "arrange the balls in a line," and so on. Refer to A.2.3 for unseen-hard examples of natural language goals. | Agent completes four novel tasks: Make-Coffee, Locked-Door-Open, Hammer, and Soccer. Example: (1) Make-Coffee: *Engage the gripper to manoeuvre the coffee cup beneath the spout of the coffee machine.* (2) Hammer: *Utilize the gripper to grasp the hammer and strike the nail at the designated spot.* |

Table 2: Detailed descriptions of the training goals and novel goals.

## A.2 Novel Goal List for CLEVR-Robot

### A.2.1 Rephrasing Goal

We use 40 different NL expressions as the novel goals generated by ChatGPT to express different target configuration. For example, if we take a goal configuration such as "red ball and blue ball", its corresponding NL instructions (i.e., eighteen NL sentence patterns) can be one of the following:

- I can't stand the red ball ahead of the blue one. Could you switch the positions of them?
- The sight of the red ball ahead of the blue one bothers me. Can we reverse their order?
- I really dislike how the red ball is positioned in front of the blue ball. Could you exchange their places?
- It annoys me to see the red ball in front of the blue ball. Can we swap them around?
- Seeing the red ball ahead of the blue ball fills me with frustration. Let's switch them.
- The placement of the red ball in front of the blue ball is something I detest. Can you flip them?
- I can't bear to see the red ball positioned in front of the blue ball. Would you mind interchanging them?
- It irks me to have the red ball come in front of the blue ball. Could we trade their positions?
- The red ball being in front of the blue ball is something I can't tolerate. Let's switch them up.
- It really bothers me that the red ball precedes the blue ball. Can we swap their positions, please?
- Move the red ball gently to the left of the blue ball.
- Slowly nudge the red ball to the left side of the blue ball.
- Push the red ball towards the left of the blue ball at a leisurely pace.
- Slide the red ball to the left of the blue ball with a gentle touch.
- Gradually maneuver the red ball to the left side of the blue ball.
- Hasten the movement of the red ball to the left of the blue ball promptly.
- Expeditiously maneuver the red ball to the left side of the blue ball.
- Swiftly propel the red ball to the left, positioning it adjacent to the blue ball.
- Urgently shift the red ball to the left of the blue ball with rapidity.
- Accelerate the motion of the red ball towards the left of the blue ball expeditiously.
- Push the red sphere ahead of the blue sphere.
- Drive the red orb in front of the blue orb.
- Launch the red ball forward, preceding the blue one.
- Catapult the red sphere ahead of the blue sphere.
- Thrust the red sphere in front of the blue sphere.
- Propel the red-colored orb forward, leading the blue-colored orb.
- Send the red ball ahead of the blue ball.
- Fling the red sphere in front of the blue sphere.
- Hurl the red ball forward, preceding the blue ball.
- Cast the red sphere ahead of the blue sphere.
- The sight of the red ball positioned to the right of the blue ball brings me joy.
- It pleases me to observe the red ball situated on the right side of the blue ball.
- Seeing the red ball to the right of the blue ball fills me with happiness.
- I feel delighted witnessing the red ball located to the right of the blue ball.
- It brings me satisfaction to see the red ball positioned to the right of the blue ball.
- I am happy to notice the red ball situated on the right-hand side of the blue ball.

- Observing the red ball to the right of the blue ball brings me contentment.
- I am pleased by the arrangement of the red ball to the right of the blue ball.
- The red ball being on the right side of the blue ball gives me a sense of satisfaction.
- I find joy in the sight of the red ball being positioned to the right of the blue ball.

### A.2.2 Unseen (Easy)

In unseen (easy) task, the agent needs to move one ball to a specific direction. The natural language goal can be one of the following:

- Move the ball backward, it's red.
- Push the red ball in reverse.
- Back up the red ball, please.
- Shift the red ball backwards.
- Can you move the red ball backwards?
- Retract the red ball, moving it backwards.
- Put the red ball in backward motion.
- Move the red ball back, not forward.
- Send the red ball backward, if you can.
- Maneuver the red ball to the rear.
- Drive the red ball backward, please.
- Pull the red ball back.
- Make the red ball move backwards.
- Shift the red ball rearward.
- Go backwards with the red ball.
- Execute a backward movement with the red ball.
- Make the red ball's position backward.
- Pull the red ball towards the back.
- Slide the red ball backwards.
- Propel the red ball backward, if possible.
- Kindly relocate the red sphere towards the left.
- Would you mind shifting the red orb to the left?
- I request that you move the red spherical object to the left.
- Could you please transfer the red ball towards the left?
- It would be appreciated if you could shift the red ball to the left.
- Please adjust the position of the red ball to the left.
- Kindly reposition the red ball to the left.
- I'd like you to move the red ball to the left, please.
- Please ensure the red ball is moved to the left.
- Could you relocate the red ball to the left?
- Please shift the red ball leftwards.
- Please slide the red ball over to the left.
- I need you to move the red ball leftward, please.
- Please nudge the red ball towards the left.
- Please push the red ball to the left.
- Would you kindly push the red ball towards the left?

- Kindly shift the red ball in the leftward direction.
- Could you move the red ball to the left side?
- It's required that you move the red ball towards the left.
- Please execute a leftward movement of the red ball.
- Kindly shift the red sphere ahead.
- Would you mind advancing the red orb?
- Can you push the red ball onward?
- Please nudge the red-colored sphere ahead.
- Kindly relocate the red-colored orb forward.
- Would you be so kind as to move the red-colored ball forward?
- Can you shift the red-colored sphere ahead?
- Please push the red-hued orb onward.
- Kindly advance the red-colored ball forward.
- Would you mind nudging the red sphere ahead?
- Can you move the red-colored ball forward?
- Please shift the red-toned orb onward.
- Kindly relocate the red-hued sphere forward.
- Would you be so kind as to push the red-colored ball forward?
- Can you nudge the red-colored sphere ahead?
- Please move the red-colored orb forward.
- Kindly advance the red-hued ball forward.
- Would you mind shifting the red sphere ahead?
- Can you push the red-colored orb onward?
- Please nudge the red-hued ball forward.
- Kindly relocate the red sphere to the starboard side.
- Move the red orb towards the right.
- Could you shift the red ball to the right?
- I request that you move the red ball to the right.
- Please shift the red ball to the right.
- Move the ball, which is red, to the right.
- Would you mind moving the ball, which is colored red, to the right?
- Kindly relocate the spherical object of red hue towards the right.
- Can you shift the ball, which happens to be red, to the right?
- I'd appreciate it if you could move the red ball to the right.
- Please adjust the position of the red ball to the right.
- Could you possibly move the red ball to the right?
- It would be great if you could move the red ball to the right.
- Kindly transfer the red-colored ball to the right.
- Move the ball that has the color red to the right.
- Would you kindly relocate the ball, specifically the red one, to the right?
- Please make the red ball move to the right.
- Can you shift the ball that's red to the right?
- Move the ball with the red hue to the right, please.
- Could you adjust the position of the ball, specifically the one that's red, to the right?

### A.2.3   Unseen (Hard)

We designed 4 types of tasks for testing KALM's performance on the completed unseen tasks: combination of two simple tasks, combination of three simple tasks, object arrangement task, and object collection task.

- NL sentence patterns used in combination of simple tasks (Using "red ball *behind* blue ball" as goal configuration):

    1. Push the red ball behind the blue ball.
    2. Move the red ball behind the blue ball.
    3. Keep the red ball behind the blue ball.
    4. Help me push the red ball behind the blue ball.
    5. Help me move the red ball behind the blue ball.
    6. Help me keep the red ball behind the blue ball.

- Combination of two simple tasks: Push the red ball behind the blue ball and move the green ball behind the purple ball.

- Combination of three simple tasks: Push the red ball behind the blue ball and move the green ball to the left of the purple ball and keep the cyan ball in front of the red ball.

- Object arrangement task

    1. Place the balls horizontally, lining them up from left to right, in the order of red, blue, green, purple, and cyan.
    2. Set up the balls in a row from left to right, with red, blue, green, purple, and cyan in sequence.
    3. Arrange the balls in a line, moving from left to right, with red, blue, green, purple, and cyan.
    4. Position the balls horizontally, organizing them from left to right, following the sequence of red, blue, green, purple, and cyan.
    5. Line up the balls horizontally, sequencing them left to right as follows: red, blue, green, purple, and cyan.
    6. Order the balls in a row from left to right, with the sequence being red, blue, green, purple, and cyan.
    7. Arrange the balls in a horizontal line, starting from the left and proceeding to the right, with red, blue, green, purple, and cyan in order.
    8. Place the balls in a row horizontally, from left to right, in the sequence: red, blue, green, purple, and cyan.
    9. Set up the balls horizontally, arranging them in the order of red, blue, green, purple, and cyan from left to right.
    10. Line up the balls horizontally, sequencing them from left to right: red, blue, green, purple, and cyan.
    11. Position the balls in a horizontal row, ordering them left to right as follows: red, blue, green, purple, and cyan.
    12. Organize the balls horizontally, moving from left to right, with the sequence being red, blue, green, purple, and cyan.
    13. Place the balls in a line horizontally, arranging them from left to right, in the following order: red, blue, green, purple, and cyan.
    14. Set up the balls in a row horizontally, starting from the left and proceeding to the right, with red, blue, green, purple, and cyan in sequence.
    15. Arrange the balls in a horizontal line, sequencing them left to right as follows: red, blue, green, purple, and cyan.
    16. Order the balls in a horizontal row from left to right, with the sequence being red, blue, green, purple, and cyan.
    17. Position the balls horizontally, organizing them in the order of red, blue, green, purple, and cyan from left to right.
    18. Line up the balls horizontally, arranging them from left to right: red, blue, green, purple, and cyan.

19. Place the balls in a horizontal row, ordering them left to right as follows: red, blue, green, purple, and cyan.
20. Set up the balls in a row horizontally, moving from left to right, with the sequence being red, blue, green, purple, and cyan.

- Object collection task

1. Position all the other balls around the green ball, considering it as the circle's focal point.
2. Use the green ball as the nucleus of the circle, arranging the rest around it.
3. Let the green ball be the anchor of the circle, and arrange the others accordingly.
4. Make the green ball the center of attention in the circle and rearrange the others accordingly.
5. Arrange all other balls around the green one, treating it as the hub of the circle.
6. Centralize the circle around the green ball, repositioning the others accordingly.
7. Focus the circle around the green ball, adjusting the positions of the others.
8. Orient the other balls around the green one, treating it as the central axis of the circle.
9. Use the green ball as the reference point for the circle's arrangement, positioning the others around it.
10. Position all the other balls around the green ball to create the circle.
11. Arrange the other balls around the green ball, making it the center of the circle.
12. Let the green ball dictate the layout of the circle, with the other balls positioned around it.
13. Create the circle with the green ball as the center, arranging the others accordingly.
14. Use the green ball as the pivot for the circle, arranging the other balls around it.
15. Organize the circle around the green ball, adjusting the positions of the other balls.
16. Centralize the arrangement of the circle around the green ball, repositioning the others.
17. Treat the green ball as the central node of the circle and arrange the other balls accordingly.
18. Position all the other balls around the green ball, with it as the focal point of the circle.
19. Arrange the circle with the green ball at the center and the others positioned around it.
20. Base the arrangement of the circle on the green ball, repositioning the others accordingly.

## A.3  Hyper-parameters

The hyper-parameters for implementing KALM are presented in Table 3. We use the same hyper-parameters, offline RL algorithms, and policy network architecture when implementing baseline methods.

Table 3: Hyper-parameters.

| Hyper-parameters | Value |
|---|---|
| Discount Factor $\gamma$ | 0.99 |
| BC Batch Size | 100 |
| BC LR | 1e-3 |
| BC Imitation Weight | 0.5 |
| CQL Batch Size | 32 |
| CQL LR | 6.25e-5 |
| CQL Conservative Weight | 10.0 |
| TD3+BC Batch Size | 256 |
| TD3+BC Actor LR | 3e-4 |
| TD3+BC Critic LR | 3e-4 |
| TD3+BC Alpha | 0.5 |
| AWAC Batch Size | 1024 |
| AWAC Actor LR | 3e-4 |
| AWAC Critic LR | 3e-4 |
| AWAC Lambda | 1.0 |
| Feature Extractor Net | $[|\mathcal{S}| + |\mathcal{L}|, 256, 256]$, ReLU |

## A.4  Prompts for LLM Supervised Fine-tuning

- Dynamics prediction: *You are an expert in identifying environmental dynamics change. Current state is [$s_t$], after executing action [$a_t$], we get next state: [ANSWER].*
- Rollout to goal translation: *Translate the state/action rollout to textual goal.\n Rollout:[ROLLOUT]\n Goal: [ANSWER].*
- Goal to rollout translation: *Translate the textual goal to state/action rollout.\n Goal:[G].\n Rollout: [ANSWER]*

Here, [ANSWER] is the content that LLM should generate.

# B   Algorithm Descriptions

Algorithm 1, 2, and 3 present the procedures of the LLM grounding, Rollout generation and Offline RL training, respectively.

---

**Algorithm 1** LLM Grounding with Supervised Fine-tuning.

---

**Input**: Offline dataset $\mathcal{D} = \{G^k, (s_0^k, a_0^k, s_1^k, a_1^k, \cdots)\}_{k=1}^K$, and pre-trained LLM $\mathcal{M}_\theta$.

1:  Construct the SFT dataset $\mathcal{D}'$ based on $\mathcal{D}$ following method described in Sec. 3.2.
2:  **while** training not complete **do**
3:      Sample a batch of data from $\mathcal{D}'$.
4:      Update $\theta$ to minimize the SFT loss, MSELoss for continuous data, CrossEntropyLoss for discrete data.
5:  **end while**
6:  **return** the fine-tuned LLM $\mathcal{M}_\theta$.

---

**Algorithm 2** Imaginary Rollout Generation.

---

**Input**: A set of predefined novel goals: $\mathcal{D} = \{G^m\}_{m=1}^M$ and the fine-tuned LLM $\mathcal{M}_\theta$.

1:  Initialize the dataset for storing imaginary rollouts $\mathcal{D}^m = \{\}$.
2:  **for** each goal $G$ in $\mathcal{D}$ **do**
3:      Sample an initial state $s_0$ randomly from the initial state distribution.
4:      Generate imaginary rollout $(s_0, a_0, s_1, a_1, \cdots)$ based on $G$ and $s_0$ with $\mathcal{M}$, using goal-oriented prompt.
5:      Append $\{G, (s_0, a_0, s_1, a_1, \cdots)\}$ to $\mathcal{D}^m$.
6:  **end for**
7:  **return** imaginary rollout dataset $\mathcal{D}^m$.

---

**Algorithm 3** Training procedure of the offline RL policy.

---

**Input**: Offline dataset $\mathcal{D} = \{G^k, (s_0^k, a_0^k, r_0^k s_1^k, a_1^k, r_1^k, \cdots)\}_{k=1}^K$, imaginary rollout dataset $\mathcal{D}^m = \{G^m, (s_0^m, a_0^m, s_1^m, a_1^m, \cdots)\}_{m=1}^M$, and text encoding model $\mathcal{M}_\phi$.

1:  Initialize the complete dataset $\mathcal{D}^c = \{\}$.
2:  **for** $\{G, (s_0, a_0, s_1, a_1, \cdots)\}$ in $\mathcal{D} \bigcup \mathcal{D}^m$ **do**
3:      Encode the natural language goal $G$, to encoding $G_f$ using $\mathcal{M}_\phi$.
4:      Update each state $s_t = [s_t, G_f]$.
5:      Compute rollout rewards $(r_0, r_1, \cdots)$ using a with reward function.
6:      Append $\{(s_0, a_0, r_0, s_1, a_1, r_1, \cdots)\}$ to $\mathcal{D}^c$.
7:  **end for**
8:  Initialize the policy $\pi$.
9:  **while** training not complete **do**
10:     Sample a batch $\{(s_i, a_i, r_i)\}_{i=1}^n$ from $\mathcal{D}^c$.
11:     Update the policy $\pi$ with offline RL algorithm.
12: **end while**
13: **return** the optimized policy $\pi$.

---

# C  Additional Results

## C.1  Training Curves of Different Methods

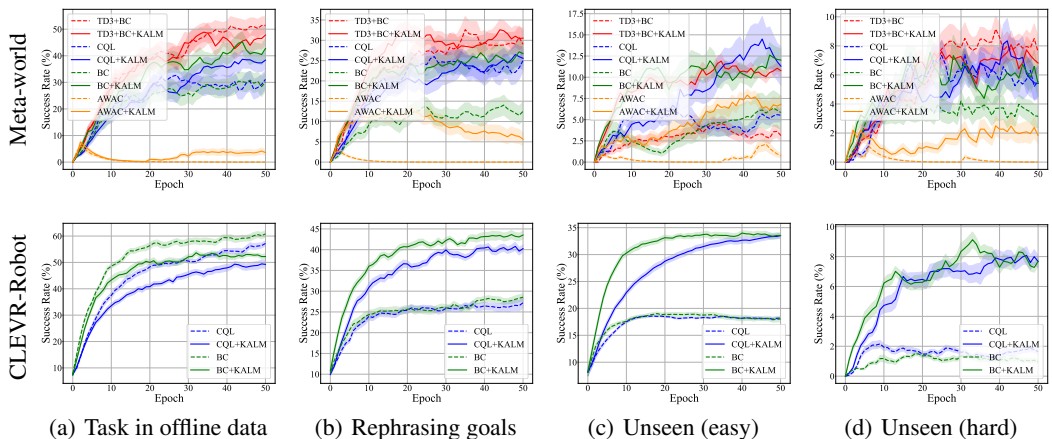

Figure 8: Success rate curves of different methods on various levels of goals. The x-axis denotes the training epochs, and the y-axis denotes the success rate for completing various natural language goals. The shaded area stands for the half standard deviation over three random seeds.

## C.2  More Examples of Generated Rollouts

The generated imaginary rollouts are important for policy optimization [60]. We present additional examples of imaginary rollouts generated by LLM in Fig. 9, 10 and 11. Overall, the generated imaginary rollouts conform to the basic physical rules of the world: LLM can correctly capture the novel goals' objects, directions and meaning.

**For CLEVR-Robot environment:** On rephrasing goal (Fig. 9(a)): LLM successfully captures the semantics contained in the goals rather than directly perceives the task goals through character matching. We also find that the performance of LLM might be sensitive to initial observation. As illustrated in Fig. 9(b), the natural language goal is easy to understand, while the LLM still fails due to the unfamiliar initial observation. On unseen (easy) tasks (Fig. 10(a)), the LLM adeptly captures the information from the goal, even if the goal is substantially different from those encountered in the training set. However, from Fig. 10(b), we find that while the system demonstrates proficiency in colour recognition, it encounters challenges in accurately discerning the direction of motion. On unseen (hard) (Fig. 11[Row 1]), although the task requirements are more complex, requiring manipulations on multiple objects (whereas the training set of LLM only contains examples about manipulations on a single object), LLM is still able to complete the task. However, as illustrated in Fig. 11[Row 2], as the difficulty of the task continues to increase, that is, when the relative positional relationships of multiple objects need to be additionally considered, LLM is no longer effective. Furthermore, although LLM did not accurately arrange the objects in a row according to the prescribed colour order, there was a clear tendency to align the objects in a row.

**For Meta-world environment:** On rephrasing goal (Fig. 9(a)): LLM can more accurately simulate simple interaction between objects. The grasped object will move with the gripper in the imaginary rollouts. As illustrated in Fig. 9(b), as interactions between objects become more complex, LLM cannot accurately model this process, resulting in the weird interaction between gripper and window. On unseen (easy) tasks (Fig. 10(a)), LLM understands the physical world relatively well. After discovering that the gripper was blocked, it adjusted and finally moved it to the target position. As illustrated in Fig. 10(b), Although LLM failed to complete the task in compliance with the laws of real physics (because of the dangling faucet), LLM did close the faucet. On unseen (hard) (Fig. 11), for a complex task which is a combination of basic tasks (first push the coffee cup, then press the coffee machine button), the LLM can only complete part of the task (pushing the coffee cup).

- Rephrasing Goal

Meta-world

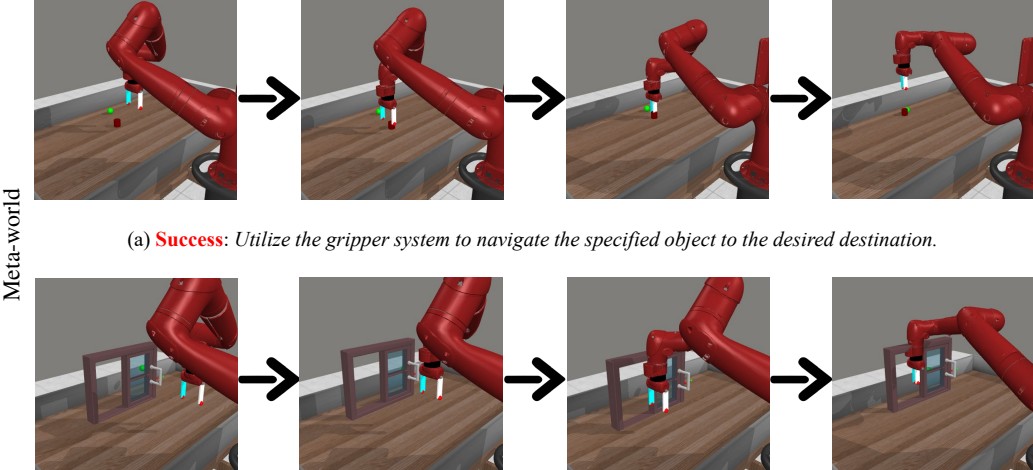

(a) **Success**: *Utilize the gripper system to navigate the specified object to the desired destination.*

(b) **Failure**: *The closed window is bothersome to me; could you please utilize the gripper to open it?*

CLEVR-Robot

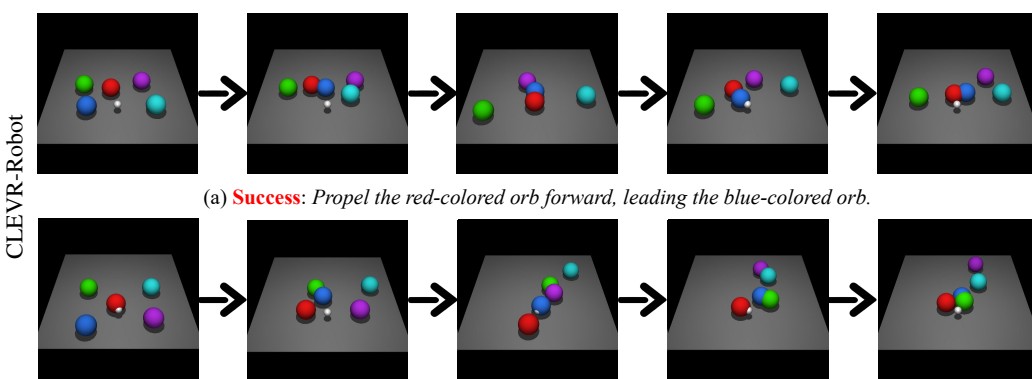

(a) **Success**: *Propel the red-colored orb forward, leading the blue-colored orb.*

(b) **Failure**: *The placement of the red ball in front of the blue ball is something I detest. Can you flip them?*

Figure 9: Additional examples of the generated rollouts for rephrasing goal tasks.

- Unseen (Easy)

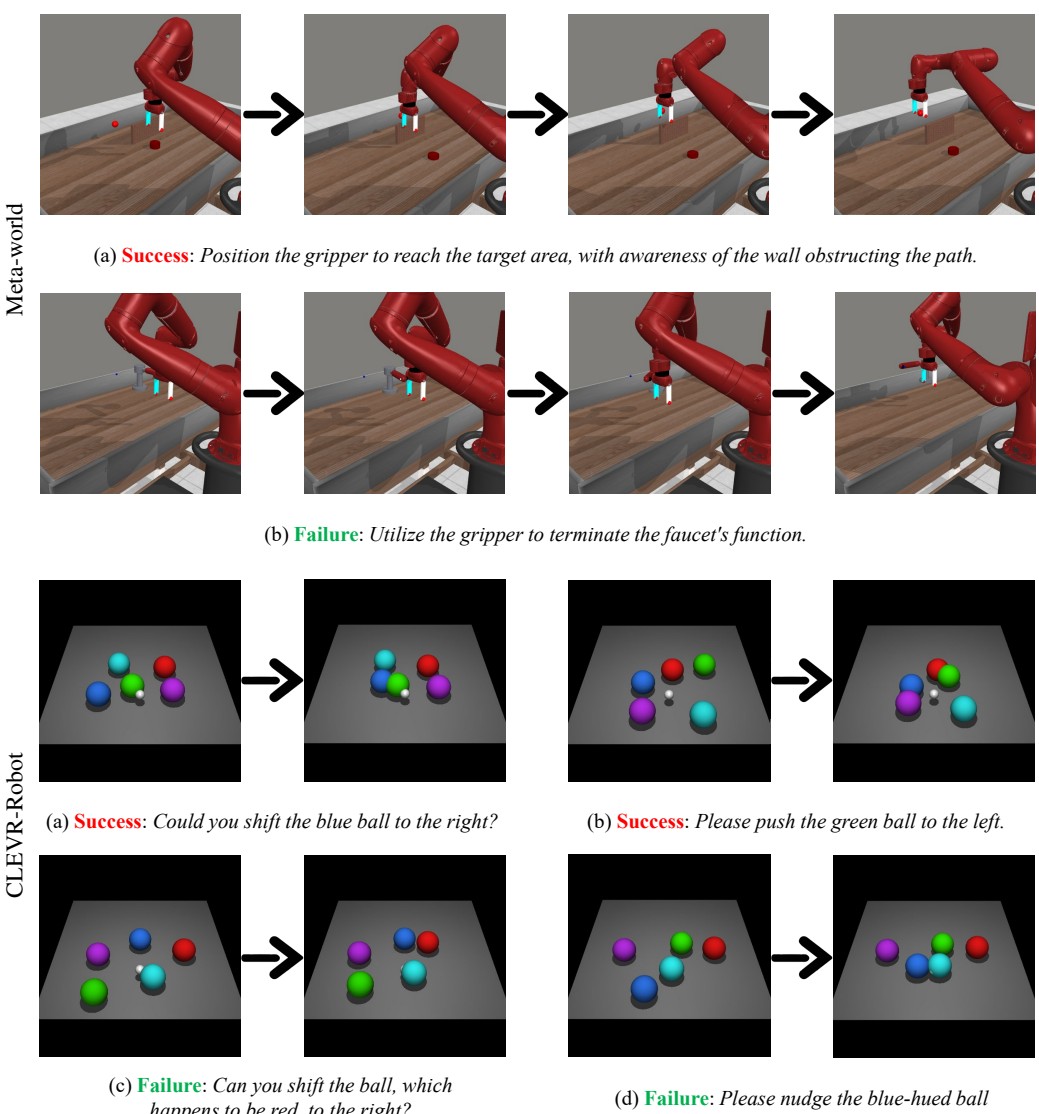

(a) **Success**: *Position the gripper to reach the target area, with awareness of the wall obstructing the path.*

(b) **Failure**: *Utilize the gripper to terminate the faucet's function.*

(a) **Success**: *Could you shift the blue ball to the right?*

(b) **Success**: *Please push the green ball to the left.*

(c) **Failure**: *Can you shift the ball, which happens to be red, to the right?*

(d) **Failure**: *Please nudge the blue-hued ball forward.*

Figure 10: Additional examples of the generated rollouts for unseen (easy) tasks.

- Unseen (Hard)

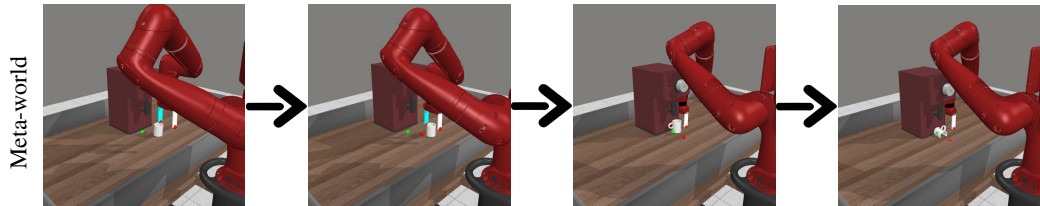

*Employ the gripper to maneuver the coffee mug into place beneath the coffee machine's spout, ready for brewing.*

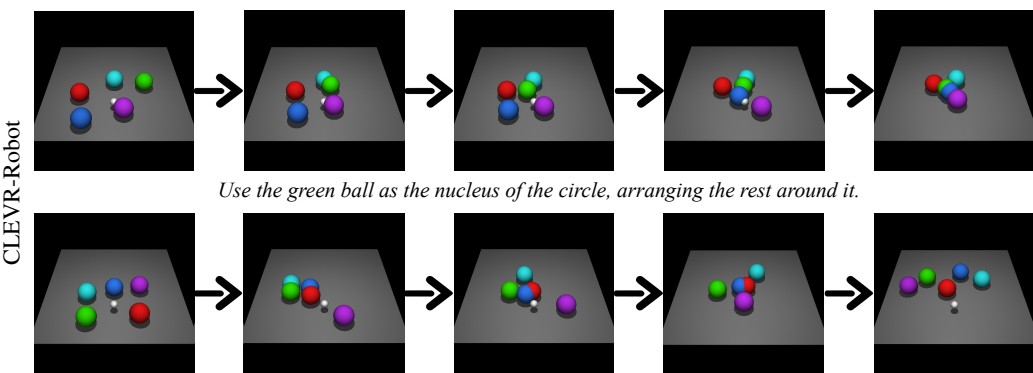

*Use the green ball as the nucleus of the circle, arranging the rest around it.*

*Position the balls horizontally, organizing them from left to right, following the sequence of red, blue, green, purple, and cyan.*

Figure 11: Additional examples of the generated rollouts for unseen (hard) tasks.

## C.3 Examples of Rollout explanations

We present a few examples of generated rollout explanations used to calculate accuracy in Figure 7. The prompt used to generate an explanation of Seen Rollouts is presented in A.4, and the prompt for Unseen Rollouts is *Suppose you are playing a gaming of five balls with different colours. Please explain the following trajectory briefly.\nTrajectory:[ROLLOUT]\nAnswer:[ANSWER]*

Success cases:

- Model output: The green ball is being moved left of the cyan ball
- Label: The green ball is pushed to the left of the cyan ball

- Model output: The cyan ball is being moved in front the purple ball
- Label: Push the purple ball in front of the cyan ball

Failure case:

- Model output: The red ball is being moved behind the cyan ball
- Label: The blue ball is pushed in behind the cyan ball

## C.4 Step Match Rate of the Generated Rollouts on CLEVR-Robot

We examine the quality of the generated rollouts on unseen (easy) tasks. At this task level, the agent aims to reposition a ball in a specified direction in one step. The quality of the generated data is quantified by measuring the generation accuracy, which is determined by the match between the generated state/action and the given goal. For instance, given the goal "Move the red ball to the left," the accuracy is calculated by checking the alignment ratio of the generated state/action with this

natural language goal. Fig. 12 shows the results of the generation accuracy of five checkpoints during the training. The results show that while action generation accuracy remains constant at approximately 30%, state generation accuracy improves as training progresses. These findings suggest that LLM works better in generating states than generating actions. Besides, the LLM can generate imaginary rollouts for novel tasks without being explicitly trained on such tasks.

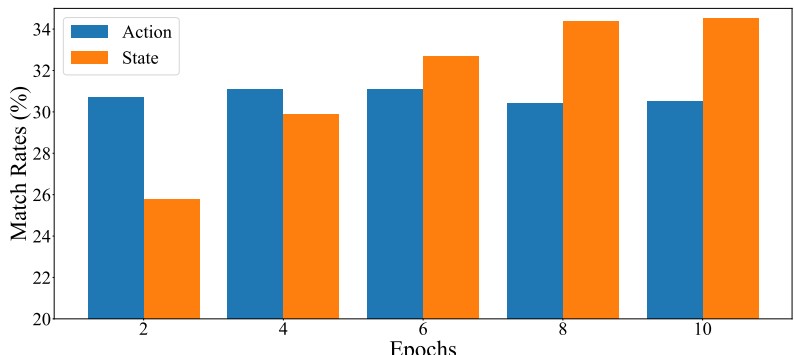

Figure 12: Single step match rate of the generated rollouts, which is assessed by examining the alignment between the states and actions generated by the LLM and the labelled goals on unseen (easy) tasks. The horizontal axis represents the training epochs, while the vertical axis is the match rate of the generated states/actions.

## C.5 Quality of the Generated Rollouts

We conduct statistics on the quality of the generated rollouts. Note that the experiments focus on the environment state in numerical vector, where each dimension in the vector has specific semantic. Therefore, the LLM would not generate rollouts containing impossible objects. Here we focus on investigating the implausible rollouts as follow topics: 1. the object is out of the table; 2. the object floats in the air; 3. implausible robotics joint pose (e.g., out of the joint rotation angle bound) and 4. exceeding dynamics limits between two steps. The statistics results are presented in Table 4 and 5.

Table 4: Ratios of various types of unrealistic transitions of the imaginary rollouts on CLEVR-Robot.

|  | Out of workbench | Exceed dynamics limits |
| --- | --- | --- |
| Rephrasing goals | 18.4 | 0.1 |
| Unseen (easy) | 0.0 | 0.2 |
| Unseen (hard) | 4.3 | 0.2 |

Table 5: Ratios of various types of unrealistic transitions of the imaginary rollouts on Meta-world.

|  | Float in the Air | Out of workbench | Implausible pose | Exceed dynamics limits |
| --- | --- | --- | --- | --- |
| Rephrasing goals | 39.5 | 18.3 | 34.0 | 44.6 |
| Unseen (easy) | 35.5 | 10.9 | 39.2 | 47.9 |
| Unseen (hard) | 22.3 | 64.9 | 31.2 | 31.1 |

The results indicate while there is a certain level of hallucination present in the outputs of the LLM, the majority of the imaginary rollouts remain within reasonable scope. Besides, the results on CLEVR-Robot demonstrate lower anomaly ratio. These hallucinations may serve as a form of domain randomization that improves the policy's robustness. Besides, KALM provides a possibility for inferring novel task states based on the generalized knowledge of LLMs, just like the human brain's imagination on the place they've never been

# D    Retrievals for Notations and Abbreviations

Tab.6 presents a list of the notations and abbreviations employed throughout this paper, serving as a convenient reference for the reader.

| Name | Meaning |
|---|---|
| **Notations** | |
| $\mathcal{D}$ | dataset |
| $\mathcal{M}$ | large language model |
| $E_T$ | token embedding layer |
| $E_S$ | state embedding layer |
| $E_A$ | action embedding layer |
| $\mathcal{G}$ | goal space |
| $G$ | goal in the goal space |
| $|\mathcal{S}|$ | dimension of observation |
| $|\mathcal{L}|$ | dimension of language model hidden state |
| **Abbreviations** | |
| KALM | Knowledgeable Agent from Language Model Rollout |
| LLM | Large Language Model |
| BC | behavior cloning algorithm |
| CQL | conservative q-learning algorithm |
| TD3 | twin delayed deep deterministic policy gradient algorithm |
| AWAC | advantage weighted actor-critic algorithm |
| MLP | multi-layer perceptrons |

Table 6: Notations and abbreviations in this paper.

# E    Broader Impact

By integrating LLMs with RL, KALM enables creating AI agents that can understand and perform complex tasks with greater efficiency and adaptability. This could enhance industries that rely on automation and intelligent systems, such as manufacturing, where robots could learn to handle new and complicated manipulation tasks without extensive reprogramming. In the realm of robotics and automation, the ability of agents to understand and perform complex tasks with a high degree of success could lead to more efficient and versatile robotic systems. These systems could be deployed in various sectors, including manufacturing, healthcare, and disaster response, where they can perform tasks that are dangerous, tedious, or beyond human capabilities, thereby improving safety, productivity, and efficiency. KALM presents an exciting advancement in the field of RL that could impact society, while we must ensure that such AI systems are handled responsibly, considering ethical implications such as privacy.

