# OpenReview forum: "KALM: Knowledgeable Agents by Offline Reinforcement Learning from Large Language Model Rollouts"
_NeurIPS.cc/2024/Conference — NeurIPS 2024 poster_

### Official Review · Reviewer_bRuq · 2024-07-07

**Soundness:** 3
**Presentation:** 3
**Contribution:** 3
**Rating:** 5
**Confidence:** 3

**Summary:**

This paper introduces a novel approach, KALM (Knowledgeable Agents from Language Model Rollouts). KALM extracts knowledge from LLMs in the form of imaginary rollouts, which agents can learn through offline RL. KALM fine-tunes the LLM to bridge the semantic gap between LLMs and RL agents. The paper demonstrates the effectiveness of KALM on robotic manipulation tasks, achieving a higher success rate than baseline methods, especially in unseen tasks.

**Strengths:**

- The idea of extracting knowledge from LLMs in the form of imaginary rollouts is interesting and useful.
- The experimental analyses are sufficient and well-conducted.

**Weaknesses:**

- Constrained by the limited context length, generating a complete trajectory seems unreliable for more complex tasks with longer trajectories. Moreover, there is a lack of details regarding the trajectory length of the experimental tasks.
- The proposed method requires the environment-built-in reward function to obtain and incorporate rewards, which may not be accessible for some other tasks. What if training the LLM to generate the rewards?

**Questions:**

- How is the dataset collected? What is the quality of the policy used to collect data?
- Why use only 6400 rollout-goal pairs to train offline RL, given there are 100,000 rollout-goal pairs available?

**Limitations:**

The authors mentioned that the generation of both state and action increases the burden on the LLM. Additionally, the current version is limited to state in the form of vector.

---

> ### Author Rebuttal · Authors · 2024-08-07
>
> Thank you for your time and valuable comments. Below, we address each of your questions and provide detailed responses to your concerns.
>
> > Q1: Constrained by the limited context length, generating a complete trajectory seems unreliable for more complex tasks with longer trajectories. Moreover, there is a lack of details regarding the trajectory length of the experimental tasks.
>
> **Comment 1:** Constrained by the limited context length, generating a complete trajectory seems unreliable for more complex tasks with longer trajectories.
>
> **Response 1:** This is a good point. We understand your concern about the trajectory generation on long-horizon tasks. We clarity that current experiments primarily demonstrate the efficacy of our proposed methodology, which extracts knowledge from LLM to facilitate low-level control. For tasks with longer trajectories, there have been established techniques verified to be effective, e.g., uncertainty estimation and branch rollout in MOPO [1], and long-range coherence in Sora [2], which is capable of generating video exceeding one minute. While these techniques are compatible with KALM, they were not employed in this study since the average trajectory length was approximately 70 timesteps (as shown below), which did not necessitate their integration. It would be interesting for future research to investigate how these techniques could be incorporated to handle longer trajectory tasks within the KALM framework.
>
> **Comment 2:** there is a lack of details regarding the trajectory length of the experimental tasks.
>
> **Response 2:** Thanks for your suggestion. We have included the details regarding the trajectory length of offline data, as shown in the following table.
>
> |     CLEVR-Robot      | Min  | Mean | Max  |
> | :------------------: | :--: | :--: | :--: |
> | Task in offline data | 2.0  | 30.8 | 50.0 |
> |   Rephrasing goals   | 1.0  | 17.2 | 50.0 |
> |    Unseen (easy)     | 1.0  | 1.0  | 1.0  |
> |    Unseen (hard)     | 1.0  | 45.1 | 50.0 |
>
> |      Meta-world      |  Min  | Mean  |  Max  |
> | :------------------: | :---: | :---: | :---: |
> | Task in offline data | 17.0  | 69.5  | 100.0 |
> |   Rephrasing goals   | 14.0  | 73.7  | 100.0 |
> |    Unseen (easy)     | 17.0  | 54.7  | 100.0 |
> |    Unseen (hard)     | 100.0 | 100.0 | 100.0 |
>
> > Q2: The proposed method requires the environment-built-in reward function to obtain and incorporate rewards, which may not be accessible for some other tasks. What if training the LLM to generate the rewards?
>
> We acknowledge that the availability of a built-in reward function can be a limitation in environments where such functions are not readily accessible or well-defined. However, we would like to emphasize that the main focus of this work is to explore the effective utilization of knowledge in LLM for low-level control, and the proposed method is not inherently dependent on the existence of such a reward function. Considering the potential absence of such functions, training LLM to generate rewards is a potential solution, while it may add burden to the LLM training. As an alternative, there have been other well-validated methods utilizing LLM to generate reward function [3,4] or dealing with sparse reward problems [5,6].
>
> > Q3: Questions about experiment setting.
>
> **Comment 1:** How is the dataset collected? What is the quality of the policy used to collect data?
>
> **Response 1:** We collect the offline dataset by utilizing expert policies specifically trained to complete natural language goal for each task. To simulate the noise in real-world data, we introduce a small portation (10%) of rollouts generated by the random policy. In this way, the dataset reflects a degree of the variability of the environment. We would incorporate these details in the experiment setting in the revised version.
>
> **Comment 2:** Why use only 6400 rollout-goal pairs to train offline RL, given there are 100,000 rollout-goal pairs available?
>
> **Response 2:** We apologize for the confusion brought by the dataset utilization. This is attributed to an issue of computation resources. The generation of imaginary rollouts is computationally expensive and time-consuming, with the capacity to generate approximately 6,400 pairs in a 24-hour period. Therefore, to prevent potential sample bias and maintain a balanced dataset, we choose an equivalent number of offline rollouts from the available 100,000 pairs.
>
> Nevertheless, we acknowledge it worth exploring to utilize all the 100,000 offline rollouts, by balancing the imaginary / offline rollout in each training batch. The experiment results on CLEVR-Robot are shown in the table:
>
> |                       | BC [6,400] | BC+KALM [6,400] | BC [100,000] | BC+KALM [100,000] |
> | :-------------------: | :--------: | :-------------: | :----------: | :---------------: |
> | Tasks in offline data |    63.1    |      56.3       |   **70.4**   |       63.4        |
> |   Rephrasing goals    |    26.3    |      44.6       |     26.3     |     **51.1**      |
> |     Unseen (easy)     |    17.5    |    **33.5**     |     16.8     |     **33.7**      |
> |     Unseen (hard)     |    1.4     |       7.2       |     1.0      |      **7.7**      |
>
> The result suggests that an increase in the amount of offline rollouts can lead to improvements in method performance. Overall, the KALM still outperforms the baseline under this setting.
>
> ## Reference
>
> [1] MOPO: Model-based Offline Policy Optimization. Tianhe Yu, et al. NeurIPS 2020.
>
> [2] Video generation models as world simulators. OpenAI team. 2024.
>
> [3] Text2Reward: Reward Shaping with Language Models for Reinforcement Learning. Tianbao Xie, et al. ICLR 2024.
>
> [4] Eureka: Human-Level Reward Design via Coding Large Language Models. Yecheng Jason Ma, et al. ICLR 2024.
>
> [5] Learning by playing solving sparse reward tasks from scratch. Riedmiller, et al. ICML 2018.
>
> [6] Overcoming exploration in reinforcement learning with demonstrations. Ashvin Nair. ICRA 2018.

---

> > ### Author Response · Authors · 2024-08-13
> >
> > Hi reviewer bRuq. We wanted to follow up to see if the response addresses your concerns. If you have any further questions, please let us know. Thank you again!

---

> > > ### Comment · Reviewer_bRuq · 2024-08-13
> > >
> > > Thank you for your detailed rebuttal. I will keep my score. I believe the paper would be further strengthened by extending it to tasks with longer horizons.

---

> ### Author Response · Authors · 2024-08-14
>
> Thanks for your response and positive assessment. While current experiments have verified the feasibility of our motivation, we believe it would be valuable supplement by evaluating KALM on more complex tasks. During the rebuttal phase, we have investigated KALM's adaptability on visual tasks. In future work, we would further explore KALM's performance on tasks with longer horizon, employing the established techniques discussed in our response to Q1.

---

### Official Review · Reviewer_JwPT · 2024-07-10

**Soundness:** 3
**Presentation:** 4
**Contribution:** 3
**Rating:** 6
**Confidence:** 5

**Summary:**

This paper presents KALM, a novel approach adopting LLMs to generate imiginary rollouts to augment the offline dataset for offline RL. Structure of LLMs is altered to handle the numeric states and actions in decision-making environments. Experiments demonstrate the imaginary rollouts benefits the offline policy learning and effectively help KALM surpass the baselines.

**Strengths:**

1. Adopting LLMs to generate imaginary rollouts is a promising and interesting direction for decision-making field, as LLMs possess extensive knowledge to help model the environment mechanism.

2. This paper is well-written and clearly describes the proposed method.

3. Empirical results validate the effectiveness of using LLMs to generate rollouts. This may further pave the way for using LLMs as world models.

**Weaknesses:**

1. Considering the close connection between this paper and model-based offline RL, I'm surprised that they are not even mentioned in the related work. Model-based offline RL methods like MOPO, MoREL and COMBO demonstrate strong performances and are also capable of generating imaginary rollouts  just like KALM. They should be discussed and included in the baselines.

2. I wonder whether the evaluation between baselines with/without KALM is fair. The policy in KALM also conditions on $G$, a hidden vector generated by BERT. I think this introduces additional information to the policy. Does the other baseline also use this information?

3. This is an open question. What do you think are the major differences between KALM and world models? Is there any possibilities to extend KALM to more complicated environments (even real-world cases)?

**Questions:**

See weaknesses.

I will increase my score to 6 upon seeing comparisons between KALM and model-based offline RL. This comparison directly demonstrates the effectiveness of using LLMs instead of traditional neural networks to generate rollouts.

**Limitations:**

Yes, limitations are discussed.

---

> ### Author Rebuttal · Authors · 2024-08-07
>
> We appreciate your insightful comments and finding our idea interesting. To address your concern, we have devoted effort to include discussion and experiment on model-based RL. Please find our response below.
>
> > Q1: Model-based offline RL methods like MOPO, MOReL and COMBO demonstrate strong performances and are also capable of generating imaginary rollouts just like KALM. They should be discussed and included in the baselines.
>
> **Comment 1:** Discussion about model-based RL methods.
>
> **Response 1:** We agree with you at this point. In our initial submission, we touch upon model-based RL (MBRL) methods (MAPLE and RedM) in Section 2.2, referring to them as *environment models*. However, we acknowledge the need for a more comprehensive discussion about prevalent MBRL methods:
>
> MBRL algorithms learn a dynamic model from offline data, which can then be used by any policy learning algorithm to recover the policy. MOPO [1] and MOReL [2] use uncertainty quantification to construct a lower bound, aiming to avoid issues like model bias and distribution shift. COMBO [3] employs both the offline dataset and model-generated rollouts to train a value function and regularize it on out-of-support state-action tuples. Despite both MBRL and KALM methods utilized generated rollouts for policy training, they are different from motivation. KALM, in particular, extracts knowledge from pre-trained LLM to build knwoledgeable agent. Leveraging the LLM's general world knowledge, we demonstrate that LLM has the potential to generate rollouts for unseen goals, extending beyond the scope of offline data and enabling the acquisition of novel skills.
>
> We would like to incorporate the discussion into the revised version.
>
> **Comment 2:** I will increase my score to 6 upon seeing comparisons between KALM and model-based offline RL. This comparison directly demonstrates the effectiveness of using LLMs instead of traditional neural networks to generate rollouts.
>
> **Response 2:**  Thanks for your suggestions. We have conducted experiments to compare KALM and two representative baselines: COMBO and MOPO, as the results in the table:
>
> | |   KALM   | COMBO | MOPO
> | :-: | :-: | :-: | :-: |
> | Task in offline data | 46.1 | 39.3  | 0.2
> |   Rephrasing goals   | 30.8 | 23.3  | 0.2
> |    Unseen (easy)  | 12.5 |  3.4  | 1.0
> | Unseen (hard)  | 7.2  |  6.9  | 1.6
>
> KALM surpasses two compared model-based offline RL methods, demonstrating the effectiveness imaginary of rollouts generated from LLM.
>
> > Q2: The policy in KALM also conditions on G, a hidden vector generated by BERT. I think this introduces additional information to the policy. Does the other baseline also use this information?
>
> In our experiments, all baseline methods utilize the BERT to convert textual goals into vector representations, ensuring a fair comparison. We appologize for the omission of this detail in the description of experiment setting. To further improve the comprehensiveness of our experiment, we have now included an addition baseline, DT [4], that utilizes both LLM and offline dataset. DT utilizes Llama-2-7b-chat-hf as the backbone policy model and offline data to training the policy. It treats decision-making as a sequence modeling problem, using a transformer architecture to predict actions based on the desired future return. In this experiment, DT trains on the offline data same as other methods, and the results are as follow:
>
> |  | KALM | DT
> | :-: | :-: | :-: |
> | Task in offline data | 46.1 | 26.4
> | Rephrasing goals   | 30.8 | 19.1
> | Unseen (easy) | 12.5 | 11.4
> | Unseen (hard) | 7.2  | 4.5
>
> The results show that KALM outperforms DT on all four types of tasks.
>
> > Q3: This is an open question. What do you think are the major differences between KALM and world models? Is there any possibilities to extend KALM to more complicated environments?
>
> **Comment 1:** What do you think are the major differences between KALM and world models?
>
> **Response 1:** Both world models and KALM train model to generate environmental rollouts, but their motivations are different: KALM leverages the embedded knowledge within pre-trained LLM, utilizing this pre-existing information to generate rollouts. Conversely, world models begin with no prior knowledge and learn from extensive datasets, constructing knowledge from foundational principles.
>
> Besides, world models and KALM utilize the offline data for different purposes. World models particularly excel in visual tasks, processing observations such as high-dimensional images. They utilize offline datasets to abstract observation features and learn environmental dynamics. On the other hand, KALM utilizes offline data to train policy and applies the offline data to bridge the gap between LLM and the environmental.
>
> **Comment 2:** Is there any possibilities to extend KALM to more complicated environments (even real-world cases)?
>
> **Response 2:** As we discuss in Section 5, we have considered extending KALM to more complex tasks, e.g., tasks with visual input. We make attempt towards this direction on Meta-world environment with visual observation (visual input is closer to the real world). We utilize a randomly initialized ViT as the vision encoder/decoder to process visual input. Figure 3 in the attached PDF file shows the example of the generated rollout, given a language goal *unseen during the training*. The generated rollout successfully captures the core information of the environment, reflecting the reasonable robotic movements tendency. This result provides evidence of the potential for extending KALM to more complicated environments.
>
> ## Reference:
>
> [1] MOPO: Model-based Offline Policy Optimization. Tianhe Yu, et al. NeurIPS 2020.
>
> [2] MOReL: Model-Based Offline Reinforcement Learning. Rahul Kidambi, et al. NeurIPS 2020.
>
> [3] COMBO: Conservative Offline Model-Based Policy Optimization. Tianhe Yu, et al. NeurIPS 2021.
>
> [4] Decision Transformer: Reinforcement Learning via Sequence Modeling. Lili Chen, et al. NeurIPS 2021.

---

> ### Comment · Reviewer_JwPT · 2024-08-08
>
> Thanks for the reply. Your experiment results demonstrate LLM-based environment models are better than previous models in MBRL, even though the generated rollouts are numeric (which, to my best of knowledge, is a weakness of current LLMs). Am I getting your conclusions right?

---

> > ### Author Response · Authors · 2024-08-08
> > **Thanks for your response**
> >
> > Thanks for your response. Your understanding is right. While LLM is primarily designed for text-based tasks, the experiments show that KALM outperforms MBRL methods after fine-tuned with numeric environment data. This improvement over traditional neural network could be attributed to the increased model capacity, which improves the representational ability. In addition, the embedded knowledge within LLM can serve as an implicit regularization, aiding in the model's ability to generalize.

---

> > > ### Comment · Reviewer_JwPT · 2024-08-09
> > >
> > > I have increased my score. Please include the comparisons and discussion about MBRL in th paper should it be accepted.

---

> ### Author Response · Authors · 2024-08-12
>
> Thank you for raising your score. We are glad the additional comparisons and discussions address your concerns. We ensure these content would be incorporated in the future version.

---

### Official Review · Reviewer_1PB3 · 2024-07-11

**Soundness:** 2
**Presentation:** 2
**Contribution:** 3
**Rating:** 5
**Confidence:** 3

**Summary:**

This paper provides KALM (Knowledgeable Agents from Language Model Rollouts) that fine-tunes a LLM (e.g., Llama-2-7B-Chat) with offline RL (e.g., CQL) for robotic manipulation tasks (e.g., CLEVR-Robot and Meta-world).

KALM mainly consists of three steps: (1) LLM grounding, (2) rollout generation, and (3) offline RL. In the LLM grounding step, KALM fine-tunes a LLM in a supervised manner on an offline dataset collected from the environment. More specifically, supervised fine-tuning involves three tasks: dynamics prediction, rollout explanation, and rollout generation. In the rollout generation step, KALM uses the fine-tuned LLM to generate imaginary rollouts with goal-oriented prompt (GOP). Here, the GOP is generated by paraphrasing the goals in the offline dataset or synthesizing new goals. Finally, in the offline RL step, KALM fine-tunes the LLM on the combination of the offline dataset and the imaginary rollout dataset.

This paper evaluates KALM on two robotic manipulation benchmarks: CLEVR-Robot and Meta-world. This paper empirically demonstrates that KALM can outperform offline RL algorithms such as CQL and BC.

**Strengths:**

S1. This paper proposes a method that uses a LLM (e.g., Llama-2-7B-Chat) to generate an imaginary rollout data in addition to an offline dataset. This approach can be seen as a kind of data augmentation. And, this approach improves the performance of an agents both in-distribution and out-of-distribution test cases. Especially, this paper demonstrates that KALM performs well in the out-of-distribution cases.

**Weaknesses:**

W1. This paper mainly compares KALM with offline RL algorithms such as CQL. However, I am not sure that this comparison is fair and reasonable. Since KALM uses imaginary rollout data in addition to an offline dataset, it may not be considered as an offline method. If it true, comparing KALM with pure offline algorithms may not be fair.

**Questions:**

Q1. When generating imaginary rollouts, does KALM interact with the environment?

Q2. Does KALM use rewards from the environment to filter out low-quality rollouts?

Q3. Can we use DPO (Direct Preference Optimization) instead of offline RL algorithms? Are there any pros and cons?

**Limitations:**

The authors adequately provides limitations of their work in Section 5 (i.e., Conclusion and Limitation).

---

> ### Author Rebuttal · Authors · 2024-08-07
>
> Thank you for your valuable comments and time in reviewing this paper. It appears there may be some confusion regarding the fairness of the experimental comparison. Below, we provide a comprehensive response to address these concerns.
>
> > Q1: Concerns about the fairness of comparison.
>
> **Comment 1:** I am not sure that this comparison is fair and reasonable. Since KALM uses imaginary rollout data in addition to an offline dataset, it may not be considered as an offline method.
>
> **Response 1:** We first clarify the fairness of the comparison: all methods in our experiments are offline methods, which *does not require online interaction* with the environment (we would discuss the environment interaction in detail in response 2 below). KALM fine-tunes a general purpose LLM with the offline data, and generates imaginary rollouts without online interaction, thus adhering to the offline setting and ensuring the fair comparison. In the initial submission, we have made a comparison between KALM and a LLM-related baseline (LLM as policy), as shown in Figure 5. To further justify the proposed method and address your concern, we add an additional baseline for comparison, Decision-transformer (DT) [1], which utilizes Llama-2-7b-chat-hf as the backbone policy model and offline data to training the policy. DT treats decision-making as a sequence modeling problem, using a transformer architecture to predict actions based on the desired future return. In this experiment, DT trains on the offline data same as other methods, and the results are as follow (as well shown in Figure 1 in the attached PDF file):
>
> |                      |   KALM   |  DT  |
> | :------------------: | :------: | :--: |
> | Task in offline data | **46.1** | 26.4 |
> |   Rephrasing goals   | **30.8** | 19.1 |
> |    Unseen (easy)     | **12.5** | 11.4 |
> |    Unseen (hard)     | **7.2**  | 4.5  |
>
> The results show that KALM outperforms DT on all four types of tasks. We would like to update the paper to incorporate the additional result.
>
> **Comment 2:** When generating imaginary rollouts, does KALM interact with the environment?
>
> **Response 2:** KAML does not require interaction with the environment, and the imaginary rollouts can be generated in pure offline manner. We appologize for any confusion brought by Figure 2, which involves an *optional* online process. For clarity, it should be noted that in our comparative experiments, KALM is implemented ***without the online procedure***. The inclusion of the optional online process in Figure 2 is intended to illustrate the method's extendibility and capability for future integration with real-time environment interactions.
>
> > Q2: Does KALM use rewards from the environment to filter out low-quality rollouts?
>
> KALM does not employs rollout filter. We use all imaginary rollouts generated by LLM. We suggest that offline RL algorithm can also learn useful knowledge from failure experience. Besides, these failure rollouts may contribute to the robustness of the resulting policy by providing a diverse range of scenarios for the model to learn from, although they are not successful rollouts. This insight is aligned with the comment proposed by Reviewer rv4h.
>
> > Q3: Can we use DPO (Direct Preference Optimization) instead of offline RL algorithms? Are there any pros and cons?
>
> DPO can not be directly applied to our problem setting, as it requires preference data annotated by human or AI and is initally designed for aligning LLMs with human value. A potential adaptation of DPO for integration with KALM is treating the offline rollout as the preferred data and the generated rollouts as the less-preferred data. However, this is still unfeasible as we do not have offline rollouts (i.e., preferred data for DPO) that correspond to unseen language goals. Note that the primary motivation of this research is to leverage the knowledge of LLMs to develop agents with enhanced knowledge, this goal is not dependent on the use of any specific offline RL optimization technique.
>
> ------
>
> We hope that our response has addressed your concern and questions satisfactorily. If you had any further concerns, we are glad for discussion.
>
> ## Reference:
>
> [1] Decision Transformer: Reinforcement Learning via Sequence Modeling. Lili Chen, et al. NeurIPS 2021.

---

> ### Author Response · Authors · 2024-08-07
> **Clarification**
>
> In addition to the above rebuttal, we would like to clarify some misunderstandings.
>
> It seems that the reviewer thought that KALM uses the LLM to learn from interacting with the environment by employing reinforcement learning (or DPO as the reviewer asked) to maximize the reward from the environment. This way could have been an ONLINE RL setting, where interactions with the environment are allowed.
>
> However, our whole setting is very close to the OFFLINE RL, where no online interactions are allowed during training, but only a fixed set of offline trajectories are available, plus a general-purpose LLM model. The LLM is fine-tuned using the offline trajectories, and is then used to generate trajectories in a purely imaginary way, just the same as language generation. The real and imaginary trajectories are then fed to offline RL methods. Therefore, comparing with previous offline RL methods that can only learn from offline trajectories is not unfair.
>
> We will revise to make our setting easier to understand, and will add more LLM-related baselines as in the responses to other reviewers.

---

> ### Author Response · Authors · 2024-08-13
>
> Hi reviewer 1PB3. As the discussion period comes to an end, we want to follow up to see if the response addresses your concerns. If you have any further questions, please let us know. Thank you again!

---

> ### Comment · Reviewer_1PB3 · 2024-08-13
> **After Author Response**
>
> Thank you for providing thoughtful responses to my questions. I could understand KALM more. Especially, I appreciate the effort that the authors have made in providing additional experiments with Decision Transformers (DT). Accordingly, I raise my rating.

---

> > ### Author Response · Authors · 2024-08-14
> >
> > We are glad that our response enhances your understanding of KALM method, and makes positive influence on your assessment. Thanks for your time and consideration!

---

### Official Review · Reviewer_rv4h · 2024-07-13

**Soundness:** 2
**Presentation:** 2
**Contribution:** 3
**Rating:** 6
**Confidence:** 3

**Summary:**

To bridge the gap between agents that can act and the vast prior knowledge that language models contain, the authors propose KALM (Knolwedgable Agents from Language Model rollouts), a method for training action-taking agents from language models. KALM is a finetuning method that enables a language model to translate between textual descriptions of goals and environment rollouts. Imaginary rollouts are then used for offline RL training. The paper demonstrates the efficacy and generalizability of the method on robotic manipulation tasks.

**Strengths:**

- To the best of my knowledge, the proposed method (translating bidirectionally between environment rollouts and textual goal descriptions, then using generated imaginary rollouts for RL training) is a novel way of grounding language models as agents for low-level control in robotic manipulation.
- The experiments seem thorough (across two environments, range of tasks) and the evaluation sets are curated to measure the intended effects (generalization). The empirical results generally support the claims from the authors that adding KALM helps with policy learning across multiple RL algorithms, and compare against relevant baselines (directly finetuning the LLM as a policy) and method components are ablated, demonstrating their importance.
- The problem setting is of significance to the robotics community, where there have been an increase in recent works looking to connect the capabilities of large foundation models with lower level controls.

**Weaknesses:**

- While the method seems to work in these simple simulated settings, the method itself seems susceptible to language model hallucinations. It would be interesting to see statistics on how often the model imagines objects / obstacles that don’t exist, or implausible rollouts in the simulator. Alternatively, it would be interesting to see that having hallucinations can actually lead to increased robustness.
- Given that the motivation is to improve generalizability via LLMs, it would be helpful to also include a comparison against RL methods that use prior knowledge from LLMS (e.g. the cited LLaRP). While the LLM as a policy method touches on this, it would support the motivations of the paper more if there were direct comparisons against methods that use LLMs for world knowledge & priors.
- There are a few grammatical and writing errors in the paper:
- Line 21: “intelligent agents to acquire such ability” → “intelligent agents to acquire such an ability/such abilities”
- Line 26: “despite they are highly similar tasks” → “despite them being highly similar tasks”
- Line 28: “general tasks in text domain” → “general tasks in text domains”

**Questions:**

Why do the authors think that the ablation study for Table 1 shows that adding components of the method actually leads to a degradation of performance on rollout accuracy on the unseen (easy) set? The paper presents an explanation that the current objectives don’t address learning action semantics, but this also applies to the unseen (hard) tasks and does not explain a *degradation* in performance.

**Limitations:**

The authors discuss the limitations of their work in the paper.

---

> ### Author Rebuttal · Authors · 2024-08-07
>
> We appreciate your highlight of the method novelty and the significance of the problem setting. Please find our response to each comment as follow.
>
> > Q1: While the method seems to work in these simple simulated settings, the method itself seems susceptible to language model hallucinations. It would be interesting to see statistics on how often the model imagines objects / obstacles that don’t exist, or implausible rollouts in the simulator. Alternatively, it would be interesting to see that having hallucinations can actually lead to increased robustness.
>
> Hallucination is indeed a central issue to consider when we use LLM. The results in Table 1 provide a general result in term of matching rate between generated rollouts and the given goal. In response to your comment, we have conducted more detailed statistics on the quality of the generated rollouts. Note that current experiments focus on the environment state in numerical vector, where each dimension in the vector has specific semantic. Thus the LLM would not generate rollouts containing impossible objects. Here we focus on investigating the implausible rollouts as follow topics: 1. the object is out of the table; 2. the object floats in the air; 3. implausible robotics joint pose (e.g., out of the joint rotation angle bound) and 4. exceeding dynamics limits between two steps. The statistics results are as follow:
>
> |   CLEVR-Robot    | Out of workbench | Exceed dynamics limits |
> | :--------------: | :---------: | :--------------------: |
> | Rephrasing goals |    18.4     |          0.1           |
> |  Unseen (easy)   |     0.0     |          0.2           |
> |  Unseen (hard)   |     4.3     |          0.2           |
>
> |    Meta-world    | Float in the Air | Out of workbench | Implausible pose | Exceed dynamics limits |
> | :--------------: | :--------------: | :---------: | :--------------: | :--------------------: |
> | Rephrasing goals |       39.5       |    18.3     |       34.0       |          44.6          |
> |  Unseen (easy)   |       35.5       |    10.9     |       39.2       |          47.9          |
> |  Unseen (hard)   |       22.3       |    64.9     |       31.2       |          31.1          |
>
> The results indicate while there is a certain level of hallucination present in the outputs of the LLM, the majority of the imaginary rollouts remain within reasonable scope. Besides, the results on CLEVR-Robot demonstrate lower anomaly ratio. These hallucinations may serve as a form of domain randomization that improves the policy's robustness.
>
> > Q2: Given that the motivation is to improve generalizability via LLMs, it would be helpful to also include a comparison against RL methods that use prior knowledge from LLMS (e.g. the cited LLaRP). While the LLM as a policy method touches on this, it would support the motivations of the paper more if there were direct comparisons against methods that use LLMs for world knowledge & priors.
>
> Good point. We agree with you that an additional baseline utilizing both LLM and offline data more would be a strong supplement to our experiments. In response to your concern, we conduct comparison with an additional baseline, Decision-transformer (DT) [1], with Llama-2-7b-chat-hf as the backbone policy model. DT treats decision-making as a sequence modeling problem, using a transformer architecture to predict actions based on the desired future return. In this experiment, DT trains on the offline data same as other methods, and the results are as follow:
>
> |                      |   KALM   |  DT  |
> | :------------------: | :------: | :--: |
> | Task in offline data | **46.1** | 26.4 |
> |   Rephrasing goals   | **30.8** | 19.1 |
> |    Unseen (easy)     | **12.5** | 11.4 |
> |    Unseen (hard)     | **7.2**  | 4.5  |
>
> The results show that KALM outperforms DT on all four types of tasks. We would like to update the paper to incorporate the additional result.
>
> > Q3: Why do the authors think that the ablation study for Table 1 shows that adding components of the method actually leads to a degradation of performance on rollout accuracy on the unseen (easy) set? The paper presents an explanation that the current objectives don’t address learning action semantics, but this also applies to the unseen (hard) tasks and does not explain a *degradation* in performance.
>
> Answer: We suggest that the performance degradation is attributed to the specific nature of the unseen (easy) task, whose objective is to predict one-step transitions given unseen language goals. To be more specifically, we would discuss the two components of LLM fine-tuning in KALM (i.e., dynamics prediction and rollout explanation) respectively. For dynamics prediction objective,  unseen (easy) task objective (predicting $a_t$ and $s_{t+1}$ given $G$ and $s_t$) shares similarity, yet diverges from the dynamics prediction objective (predict $s_{t+1}$ given $s_t$ and $a_t$). This difference introduces a potential conflict in the LLM SFT. This is evidenced by the empirical results that KALM w/o dynamics prediction achieves highest rollout accuracy for unseen (easy) task. For rollout explanation objective, it focuses on giving explanation on a long rollout sequences. While this objective enriches the model's capability to provide coherence over temporal sequence, it may inadvertently detract from the model's ability to capture the immediate logic of transitions between adjacent two steps.
>
> We would like to update the paper to reflect the above discussion about the experiment results.
>
> > Q4: There are a few grammatical and writing errors in the paper.
>
> We have fixed your mentioned typos and would make a more thorough check to fix the grammatical and writing errors.
>
> ------
> We believe the experiment and the result analysis have been clearly enhanced based on your c comments. If you had any further concerns, please let us know.
>
> ## Reference:
>
> [1] Decision Transformer: Reinforcement Learning via Sequence Modeling. Lili Chen, et al. NeurIPS 2021.

---

> > ### Comment · Reviewer_rv4h · 2024-08-12
> >
> > I thank the authors for responding to my questions and concerns. I will be keeping my score.

---

> > > ### Author Response · Authors · 2024-08-13
> > > **Thanks for your response**
> > >
> > > We appreciate the reviewer's acknowledgment of our responses. Thank you for your valuable time and comments.

---

### Author Rebuttal · Authors · 2024-08-07

We express our sincere thanks to the reviewers and chairs for their valuable time and constructive feedback on this paper. We are encouraged to note that reviewers acknowledge this paper's novelty (Reviewer rv4h,JwPT,bRuq), its contribution to community (Reviewer rv4h,JwPT,bRuq) and the sufficiency/performance of the experiments (Reviewer rv4h,1PB3,JwPT,bRuq). Below we summarize the major concerns raised by reviewers and our corresponding response.

1. Requiring additional baselines (Reviewer rv4h,JwPT): we have incorporated additional baseline methods. Specifically, we have included decision-transformer that leverage both LLM and offline datasets, as well as model-based offline RL approaches.
2. Fairness of the comparison (Reviewer 1PB3): we clarify that all methods in the experiments (including KALM and baselines) are under offline setting, where no online interactions are allowed during training, and only a fixed set of offline trajectories are available for learning. Therefore, the comparison is not unfair. We have added more LLM-related baselines (suggested by Reviewer rv4h) to further justify the proposed method.
3. Extending to more complicated tasks (Reviewer JwPT,bRuq): we elaborate on the extendibility of KALM framework to more challenging setting, such as tasks with longer trajectories, absent reward function or visual inputs. We have also verified the extendibility by conducting experiment on more challenging task with visual observation. Please refer to Figure 3 in for the results.

We believe that the experiments are thorough and this paper makes significant contributions to research community, as there have been an increase in connecting the capabilities of large foundation models with lower level controls (as suggested by Reviewer rv4h). We have provided detailed responses to each reviewer's comments below, and we *are eager to* receive the reviewers' feedback! Please let us know if there are any further concerns, and we are available and glad to discuss.

------
Best wishes,

Authors of Submission #502

---

### Decision · Program_Chairs · 2024-09-25

**Decision:**

Accept (poster)

**Comment:**

Reviewers agreed that the method is both new and useful. The method is also easily adapted to other environments and will likely be generally useful for many NeurIPS readers. In particular, its ability to outperform alternatives like MOPO and COMBO which also have rollouts but are not LLM-based is interesting. In the future one might imagine some hybrid between capturing different aspects of the environment.

The authors did a good job addressing reviewer concerns about other baselines, which they should incorporate into the final manuscript. Similarly for the vision-based experiments and the more difficult settings that reviewers asked for. This only makes the paper stronger and engages with a wider literature.